

# An emerging pathway of Atlantic Water to the Barents Sea through the Svalbard Archipelago: drivers and variability

Kjersti Kalhagen[1], Ragnheid Skogseth[1], Till M. Baumann[2,3], Eva Falck[1], and Ilker Fer[2,1]

[1]Department of Arctic Geophysics, University Centre in Svalbard, Longyearbyen, Svalbard
[2]Geophysical Institute, University of Bergen, and Bjerknes Centre for Climate Research, Bergen, Norway
[3]Institute of Marine Research, Bergen, Norway

**Correspondence:** Kjersti Kalhagen (kjerstik@unis.no)

**Abstract.** The Barents Sea, an important component of the Arctic Ocean, is experiencing shifts in ocean currents, stratification, sea-ice variability, and marine ecosystems. Inflowing Atlantic Water (AW) is known to be a key driver of change. Although AW predominantly enters the Barents Sea via the Barents Sea Opening, other pathways exist but remain relatively unexplored. Summer climatology fields of temperature in the last century compared to 2000–2019 indicate warming in the trench Storfjor-
drenna and the shallow banks Hopenbanken and Storfjordbanken in the Svalbard Archipelago, and shoaling of AW, extending further into the "channel" between Edgeøya and Hopen islands. This region emerges as a pathway of AW into the northwestern Barents Sea. One year-long records from a mooring deployed between September 2018 and November 2019 at a saddle in this channel, show the flow of Atlantic-origin waters into the Arctic domain of the northwestern Barents Sea. The average current is directed eastward, into the Barents Sea, but is dominated by large variability throughout the year. Here, we investigate this
variability on time scales from hours to months. Wind forcing mediates the currents and the water and heat exchange through the channel through geostrophic adjustment to Ekman transport. The main drivers for the AW inflow and the cross-saddle transport of positive temperature anomalies are persistent strong semidiurnal tidal currents, intermittent wind-forced events, and wintertime warm water intrusions forced by upstream conditions. We propose that similar topographic constraints where the Polar Front acts like a barrier may become more important for AW inflow and heat exchange in the future. The ongoing
warming and possible shoaling of AW together with changes in the large-scale weather patterns would likely increase inflow and heat transport through the processes identified in this study.

## 1   Introduction

The shallow Barents Sea (Fig. 1) plays a crucial role in the Arctic climate system, being one of the two main gateways for Atlantic Water (AW) to enter the Arctic Ocean (e.g. Schauer et al., 2002; Smedsrud et al., 2013). Recent "Atlantification" in
the Arctic Ocean (Årthun et al., 2012; Polyakov et al., 2017), characterised by ocean warming, weakening stratification, and declining winter sea ice (Shu et al., 2021), is partly fed by AW circulating through the Barents Sea (e.g., Asbjørnsen et al., 2020). The largest volume of AW enters the Barents Sea through the Barents Sea Opening (BSO, Fig. 1). In addition, AW can enter the northwestern Barents Sea through the Northern Barents Sea Opening (Lind and Ingvaldsen, 2012; Lundesgaard et al., 2022), from the branch that enters the Arctic Ocean through Fram Strait and flows north of Svalbard (e.g., Lind and



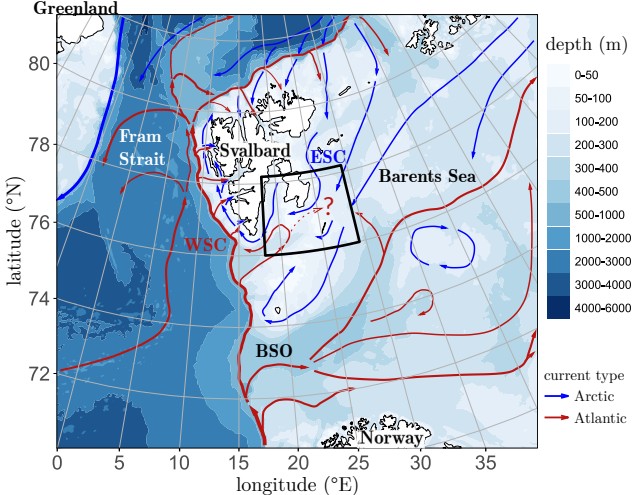

**Figure 1.** Bathymetry in the western Barents Sea and Fram Strait (IBCAO-3, Jakobsson et al. (2012)) and general circulation (arrows, Vihtakari et al. (2019), adapted from Eriksen et al. (2018)). The West Spitsbergen Current (WSC), the East Spitsbergen Current (ESC) and the Barents Sea Opening (BSO) are marked. The black rectangle depicts the area shown in the detailed map in Fig. 2.

Ingvaldsen, 2012). As AW flows through the Barents Sea, it is modified and transformed due to intense cooling through heat loss to the atmosphere (Häkkinen and Cavalieri, 1989; Årthun and Schrum, 2010), brine release from sea-ice growth (e.g., Schauer et al., 2002), and mixing with the cooler and fresher Arctic Water (Loeng, 1991). The Barents Sea has been termed a "cooling machine" (Smedsrud et al., 2013) whereby AW is transformed to Barents Sea Water. It is an important area for dense water production (Knipowitsch, 1905; Midttun, 1985; Schauer et al., 2002) and a source of intermediate waters to the Arctic
Ocean (Rudels et al., 1994; Schauer et al., 1997, 2002).

Large-scale environmental changes have been observed in the Barents Sea over the past decades (Oziel et al., 2016; Skagseth et al., 2020; Isaksen et al., 2022; Smedsrud et al., 2022). The loss of winter sea ice in the Arctic in recent times has been most pronounced in the Barents Sea (Onarheim and Årthun, 2017; Rieke et al., 2023), and the Barents Sea is the first sea of the Arctic Ocean projected to be ice-free year-round (Årthun et al., 2021). Increased heat transport through the BSO is an important driver
of the sea ice loss, warming, and Atlantification in the Barents Sea (Årthun et al., 2012; Asbjørnsen et al., 2020). Increasing air temperatures also drive sea ice loss. In the northern Barents Sea, the surface air temperatures have increased twice as fast as the Arctic as a whole (Isaksen et al., 2022). Decreased import of sea ice into the northern Barents Sea is another factor in the increased sea ice loss (Lind et al., 2018; Ingvaldsen et al., 2021), resulting in a decrease in the freshwater input and therefore a weakening of the stratification (Lind et al., 2018). The weakening stratification facilitates enhanced vertical mixing
and increased heat fluxes which may further inhibit sea ice formation (Lind et al., 2018). Further hydrographic changes in the Barents Sea include increased salinity and a diminishing presence of Arctic Water (Lind et al., 2018), changes in the Polar Front (Barton et al., 2018), which marks the boundary between waters of Atlantic and Arctic origins, and a poleward shift of the region where the cooling machine is efficient (Shu et al., 2021).



Understanding the Atlantification of the Barents Sea is important for assessing the consequences for the ecosystem (Gerland
et al., 2023), such as increased production, boreal species expanding northward, a reduction of the ice-associated ecosystem
compartment, and increased connectivity in the food web (Ingvaldsen et al., 2021). The Barents Sea net primary production has
increased substantially in the last two decades (Dalpadado et al., 2020; Lewis et al., 2020). Boreal zooplankton has expanded
northwards due to a larger area being thermally favourable (Geoffroy et al., 2018), while some Arctic zooplankton species
have retreated (Eriksen et al., 2017). The northward expansion of Atlantic zooplankton species is anticipated to increase in the
future, with possible impacts on the Arctic zooplankton communities (Wold et al., 2023). Changes in biomass and distribution
are also observed higher up in the food web (Gerland et al., 2023, and the references therein).

Although BSO is the main gateway of AW into the Barents Sea, other inflow zones such as the Northern Barents Sea Opening
can be of importance. Also, as West Spitsbergen Current (WSC) transports AW northward towards Fram Strait, some AW is
directed into Storfjordrenna in the Svalbard Archipelago (Fig. 2), north of the BSO and Spitsbergenbanken. In Storfjordrenna,
AW flows cyclonically, guided by topography, with shallow areas to its right (Vivier et al., 2023).

Storfjordrenna has had a positive sea surface temperature (SST) trend from 1982 to 2020 and the largest SST increase in the
Barents Sea during the period from 1995 to 2007 (Bayoumy et al., 2022b), due to the increased inflow of AW following shal-
lower isobaths than earlier. Following a year of record-low sea ice cover, the surface layers in Storfjorden and Storfjordrenna
were replaced by warmer and more saline Arctic Water during summer 2016 (Vivier et al., 2023). Summer of 2016 had the
warmest and longest-lasting marine heatwave in the Barents Sea (Bayoumy et al., 2022a).

Northeast of Storfjordrenna, on the Arctic side of the Polar Front, lies the Olga Basin (Fig. 2). Here, the Arctic Water layer
is the coldest and thickest compared to the rest of the northern Barents Sea (Lind and Ingvaldsen, 2012). Arctic Water enters
through the gaps and troughs in the northern Barents Sea (Dickson et al., 1970; Loeng, 1991) and circulates with the East
Spitsbergen Current (Quadfasel et al., 1988; Loeng, 1991). In addition to Arctic Water, AW is transported into the Olga Basin
at depth through the gateways in the north (Lind and Ingvaldsen, 2012; Lundesgaard et al., 2022) and to a smaller degree over
the $\approx 200\,\mathrm{m}$ deep saddle separating the Olga Basin from Hopendjupet to the south (Kolås et al., 2023; Lind and Ingvaldsen,
2012) (Fig. 2).

The channel between Edgeøya and Hopen islands (Fig. 2) separates the increasingly warm and shoaling AW in Storfjor-
drenna from the Olga Basin, the Arctic domain in the northwestern Barents Sea. The region is characterized by strong tidal
currents (Skogseth et al., 2013) and tidally generated residual flow (e.g., Kowalik and Marchenko, 2023). Here, we investigate
the physical processes that drive and mediate the inflow of Atlantic-origin water masses into the Arctic domain in the Olga
basin in the northwestern Barents Sea. This inflow of warm water into the Barents Sea affects the deep basin stratification, and
the sea ice growth and melt. We propose that the study site may be a future pathway for AW into the Arctic domain of the
Barents Sea, and identify and discuss the variability of the currents and heat exchange on time scales from hours to months.



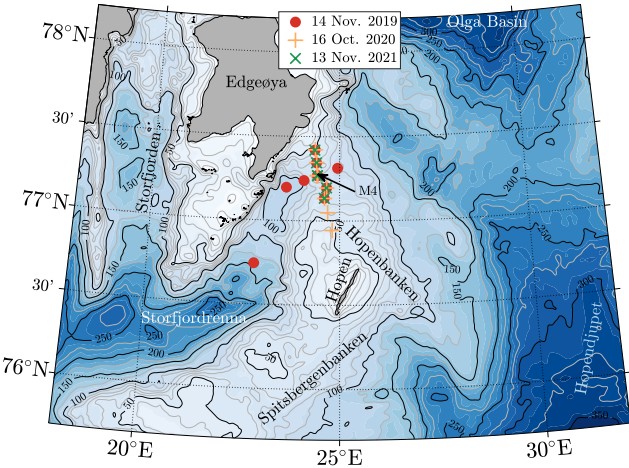

**Figure 2.** Bathymetry in the mooring area (IBCAO-4, Jakobsson et al. (2020). The mooring M4 (black arrow) is located on the saddle on Hopenbanken between the trench Storfjordrenna and the interior of the northwestern Barents Sea. The saddle is bounded to the north and south by the islands Edgeøya and Hopen, respectively. Hydrographic stations from 14 November 2019, 16 October 2020, and 13 November 2021 are marked (see legend).

## 2 Data and Methods

### 2.1 Data

In this study, historical hydrographic data, recent hydrographic transects during late autumn, and a year-long hydrographic and current time series from a mooring were used. The mooring was deployed on the 70 meter deep saddle on Hopenbanken between Storfjordrenna and the interior of the Barents Sea (Fig. 2).

The historical data include hydrographic profiles extracted from UNIS Hydrographic Dataset (UNIS HD; Skogseth et al. (2019)) covering the vicinity of the mooring area over the period 1930–2019.

The recent hydrographic transects were collected on 14 November 2019 onboard *Kronprins Haakon* (Sundfjord and Renner, 2021), on 16 October 2020 onboard *G. O. Sars* (Fer et al., 2021), and on 13 November 2021 onboard *Kronprins Haakon* (Renner and Sundfjord, 2022).

Temperature, salinity, and ocean currents were measured using instruments on the mooring from 29 September 2018 to 14 November 2019, located at $77° 16.116'$ N, $24° 24.402'$ E (Fig. 2). The mooring was a trawl-proof bottom frame and consisted of one Conductivity, Temperature, Depth instrument (CTD, an unpumped Seabird Microcat SBE37-SM) and one Acoustic Doppler current profiler (ADCP, a Nortek Signature 250) placed inside the frame with transducers pointing upward. The Microcat recorded temperature, salinity, and pressure near the bottom at about 68 m, for the whole period at a sampling interval of 15 min. The ADCP profiled current speed and direction through the whole water column with 25 cells of 3 m thickness at an averaging interval of 20 min. All data from the mooring were interpolated onto a common hourly time vector before analysis.





The velocity direction was corrected for magnetic declination of $15°$. The ADCP compass was further checked by comparing the tidal ellipses obtained from the observed current velocity with those from the Arctic Ocean 5 km Inverse Model 2018 (Arc5km2018) (Erofeeva and Egbert, 2020). For the $M_2$ constituent, the ellipse inclinations from the Arc5km2018 and the

depth-average measured current were approximately the same. For the other significant semidiurnal constituents $S_2$, $N_2$, $K_2$, the ellipse properties from the measured current were in fairly good agreement with Arc5km2018. Since $M_2$ is the dominant constituent in the study area, no further compass corrections were found necessary. The current velocity profiles were then referenced to depth below the surface to account for sea level variations that were detected by the altimeters of the ADCP.

Wind speed and direction in the region are taken from the ERA5 reanalyses (hourly data on single levels, with a spatial

resolution of $0.25°$, Hersbach et al. (2020)), while sea surface temperature (SST) and sea ice concentration (SIC) are taken from the Global Ocean OSTIA Sea Surface Temperature and Sea Ice Reprocessed product (daily data with a spatial resolution of $0.05°$, OSTIA (2023); Good et al. (2020)).

All data used in the study are openly available as detailed in the data availability section.

## 2.2 Data analysis methods

The historical hydrographic data (Skogseth et al., 2019) have been optimally interpolated onto horizontal and vertical climatological sections to compare the two recent decades (2000–2019) to the previous century (1930–2000). Due to temporally and spatially sparse data coverage, only summer (July–October mean) sections were created. A description of gridding and interpolation is given in the Appendix A. Data coverage and standard deviations for the horizontal climatological maps are shown in Fig. A2 and Fig. A1, and for the vertical climatological sections in Fig. A4 and Fig. A3.

The temperature and practical salinity from the recent hydrographic transects, from the mooring, and the optimally interpolated climatological sections were converted to Conservative Temperature and Absolute Salinity following TEOS10 (Mcdougall and Krzysik, 2015). In the following, we use the subscript $b$ (for bottom) to refer to temperature $T_b$ and salinity $S_b$ near the seafloor.

At the mooring location, the coordinate system was rotated with $-42°$ to align with the long-term mean flow along the

isobaths. The rotated velocity components are denoted as $u_R$ and $v_R$ for the along- (roughly southeast) and across-isobath (roughly northeast) directions. In the analysis of variability associated with selected time scales, we apply another rotation of the velocity components denoted as $u_r$ and $v_r$. These are the components along and across the direction of the highest variance in the frequency bands associated with weather and lower-frequency activity. This coordinate system is rotated with $-28°$ so that $u_r$ is directed roughly toward east-southeast.

Cartesian and rotary spectral components were estimated following the multitaper method from Percival and Walden (1993) with Slepian data tapers (e.g., Slepian, 1978; Thomson, 1982). To analyse the time-variability of the spectral components, wavelet transforms following Lilly and Olhede (2009) with generalised Morse wavelets (Olhede and Walden, 2002) were used. The values for the parameters that define the Morse wavelets were chosen to be $\gamma = 3$ for symmetric wavelets and $\beta = 5$ for a reasonable time and frequency resolution.





The time series of ocean currents and temperature were filtered for different aspects of the analysis. To describe the background conditions and low-frequency variability, we used a low-pass Butterworth filter of order 6 and a cutoff frequency corresponding to 10 days. To study the fluctuations in frequency bands associated with semidiurnal tides, "weather-band" processes, and lower-frequency activity, the time series were band-pass filtered. We chose cutoff frequencies corresponding to the periods ten hours to 14 hours for the semidiurnal tidal band, 28 hours to 10 days for weather-band processes, and seven days to six weeks for lower-frequency activity. We used a Butterworth filter as sharp as possible while still ensuring stability for each frequency band, order 5 for semidiurnal tides and weather-band, and order 3 for lower-frequency activity.

Eddy temperature fluxes were calculated as $\overline{u_r' T'}$ and $\overline{v_r' T'}$, where $u_r'$ and $v_r'$ are the velocity component fluctuations along ($-28°$ off east) and across ($+62°$ off east) the direction of highest variance in the weather band. Fluctuations (denoted by primes) were obtained by band-pass filtering the time series. Velocity data are depth-averaged over the bottom half of the water column. The overline denotes a 30-day moving average.

Wind stress was calculated as $(\tau_x, \tau_y) = \rho_{air} C_D |\mathbf{u}_{10\,m}| (u_{10\,m}, v_{10\,m})$, where $\rho_{air} = 1.25 \, \mathrm{kg\,m^{-3}}$ is the density of air, $\mathbf{u}_{10\,m} = (u_{10\,m}, v_{10\,m})$ is the wind vector with the eastward and northward components of wind velocity at 10 m from ERA5, $|\mathbf{u}_{10\,m}|$ is its magnitude, and $C_D$ is the drag coefficient. We used $C_D$ dependent on the sea ice concentration following Lüpkes and Birnbaum (2005). Ekman transport is calculated from the wind stress using $(U_E, V_E) = \frac{1}{\rho_0 f}(\tau_x, -\tau_y)$, where $\rho_0 = 1027 \, \mathrm{kg\,m^{-3}}$ is the reference density of sea water and $f$ is the Coriolis parameter.

Time series of SST, SIC, wind velocity, and wind stress at the mooring location, were obtained by spatially interpolating onto the mooring location from the original grids, i.e. from $0.05°$ for SST and SIC and from $0.25°$ for wind velocity.

## 3 Results

### 3.1 Long-term changes in the hydrographic environment in Storfjordrenna and on Hopenbanken

The hydrographic environment in the trench Storfjordrenna and the shallow banks Hopenbanken and Storfjordbanken (Fig. 2) has undergone warming over the past two decades. Summer climatology maps of the depth-averaged temperature in 1930–2000 (Fig. 3a) and in 2000–2019 (Fig. 3b) indicate warming over most of the area. Observed increase in average temperature exceeds $1°C$ over large parts of Hopenbanken and Storfjordbanken (Fig. 3c). The warming is observed both at the surface and at depth (not shown). The $34.7 \, \mathrm{g\,kg^{-1}}$ isohaline has extended further northeast in 2000–2019 relative to 1930–2000 (Fig. 3a–b).

The vertical climatological section from Storfjordrenna, across the saddle on Hopenbanken, and into the Olga Basin (Fig. 4) shows that waters of Atlantic origin have shoaled in the water column and now reach further east towards the saddle. The warming has occurred through the water column and the saddle area has warmed by $1\,°C$ to $2\,°C$. Additionally, salinity has increased by $0.05 \, \mathrm{g\,kg^{-1}}$ to $0.35 \, \mathrm{g\,kg^{-1}}$ over intermediate depths in Storfjordrenna (not shown).



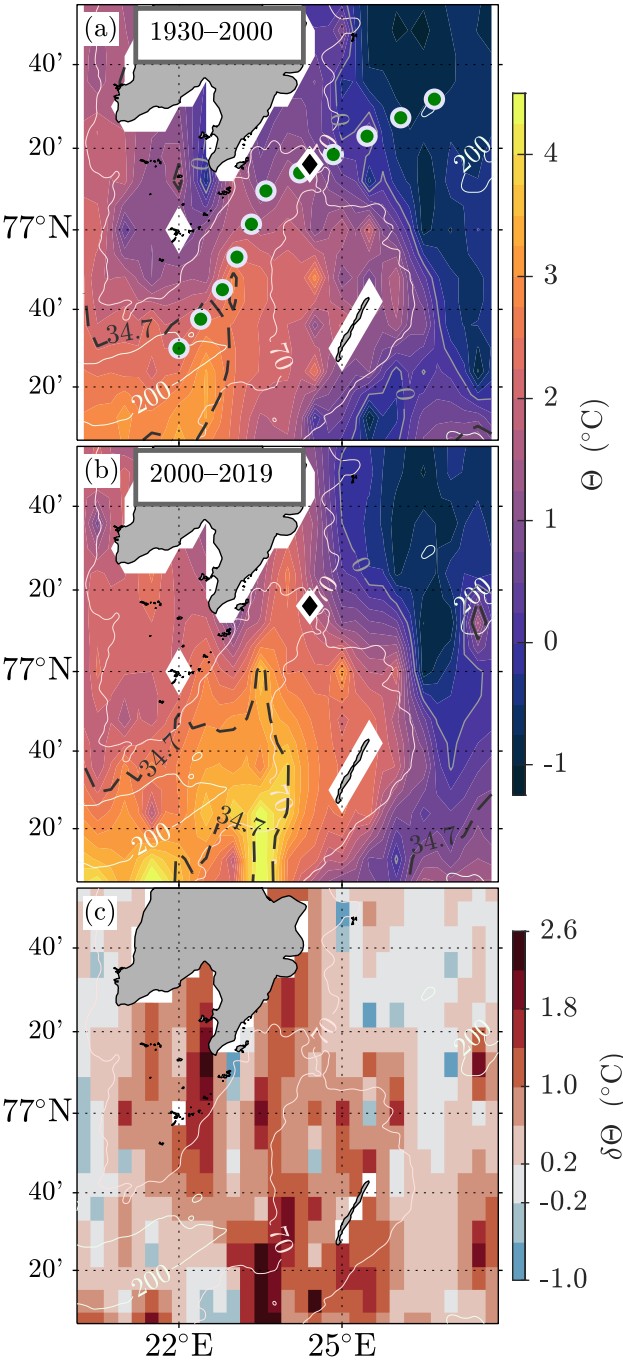

**Figure 3.** Climatological maps of depth-averaged Conservative Temperature $\Theta$ ($^\circ$C) for a) 1930–2000, b) 2000–2019, and c) their difference (1930–2000 subtracted from 2000–2019). In (a–b) the $0\,^\circ$C isotherm (light gray) and the $34.7\,\mathrm{g\,kg^{-1}}$ isohaline (black dashed) are shown for reference. In a), the vertical section in Fig. 4 is shown as green dots and the mooring position as a black diamond. The $70\,\mathrm{m}$ and $200\,\mathrm{m}$ isobaths are shown in white contours.





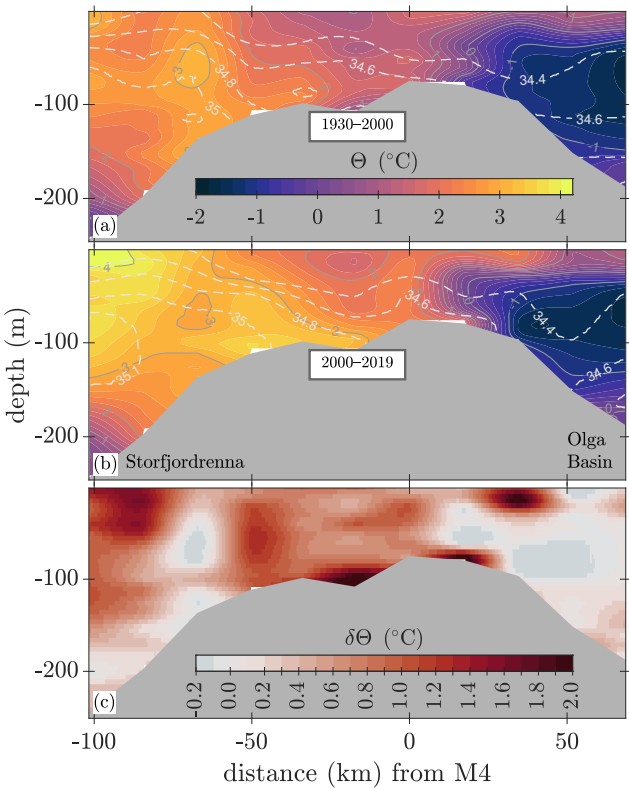

**Figure 4.** Climatological vertical hydrographic sections from Storfjordrenna, across the saddle, and into the Olga Basin for a) 1930–2000, b) 2000–2019, and c) difference (1930–2000 subtracted from 2000–2019). a–b) Conservative Temperature $\Theta$ (°C) in colours and a weak gray contour for every whole degree. The $34.4\,\mathrm{g\,kg^{-1}}$, $34.6\,\mathrm{g\,kg^{-1}}$, $34.8\,\mathrm{g\,kg^{-1}}$, $35.0\,\mathrm{g\,kg^{-1}}$, and $35.1\,\mathrm{g\,kg^{-1}}$ isohalines are shown as white dashed curves. c) Difference of Conservative Temperature $\delta\Theta$ (°C) in colours. Horizontal axis shows the distance to the mooring position, with positive values towards the Olga Basin. The location of the section is indicated with green dots in Fig. 3a.

## 3.2 Late-autumn hydrographic transects indicate large variability

Recent hydrographic transects covering the saddle region collected in November 2019 (mooring recovery cruise), October 2020, and November 2021 (Fig. 5) show that the mooring is in an area with thermal and haline gradients, with local temperature and salinity maxima near the mooring position in late autumn.

In November 2019, the transect across saddle was cold and relatively fresh but had remnants of heat (Fig. 5a). The warmest water of $-1.2\,°\mathrm{C}$ was found at the mooring position in the lower half of the water column. In the northern side of the transect, 160 the temperature was near the freezing point with relatively high salinity (Fig. 5a). At this time, AW was present in Storfjordrenna and close to the mooring (not shown): the $2\,°\mathrm{C}$ and $0\,°\mathrm{C}$ isotherms reached $50\,\mathrm{km}$ and $20\,\mathrm{km}$, respectively, to the mooring location. Also relatively warm water with $0\,°\mathrm{C}$ to $0.5\,°\mathrm{C}$ was situated at $80\,\mathrm{m}$ depth $10\,\mathrm{km}$ to the west of the mooring.





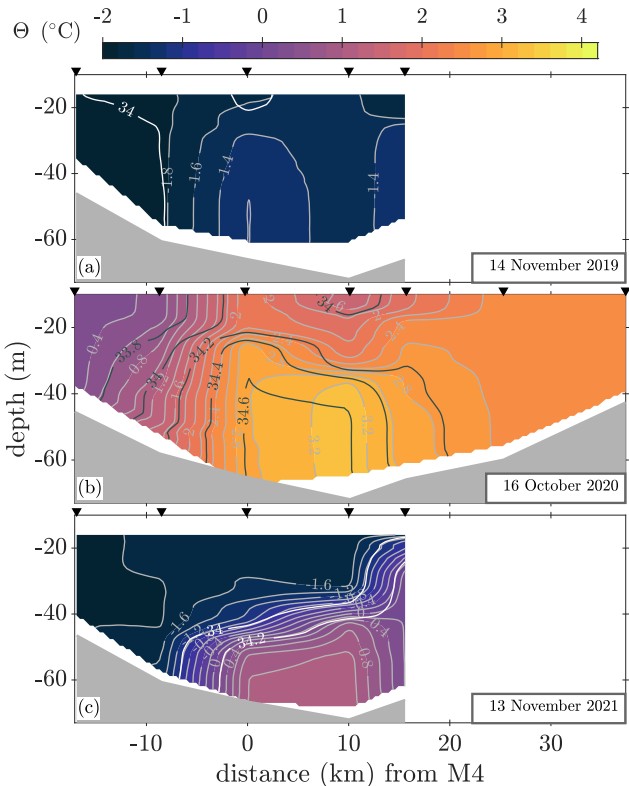

**Figure 5.** Hydrographic transects along the saddle between Storfjordrenna and the Barents Sea in a) 14 November 2019, b) 16 October 2020, and c) 13 November 2021 with Conservative Temperature in colour (contours every $0.25\,°C$ and Absolute Salinity in white (a, c) and black (b) contours (every $0.2\,g\,kg^{-1}$). The northern part of the saddle (close to Edgeøya) is to the left, and the southern part (towards Hopen) is to the right. The horizontal axis shows the distance to the mooring position M4, with positive values towards the south.

In October 2020, warm and saline water of Atlantic origin occupied the lower half of the water column (Fig. 5b). The warmest water at $3.3\,°C$ and a salinity of $34.6\,g\,kg^{-1}$ was located $10\,km$ south of the mooring position. Water warmer than

$2\,°C$ reached the surface at the mooring position and between $15\,km$ and $37\,km$ south of the mooring position. The coolest and least saline water was found on the northern side of the transect. A station $60\,km$ to the northeast showed water at $1\,°C$ and salinity $35.0\,g\,kg^{-1}$ below $150\,m$ depth (not shown).

In November 2021, the temperature maximum of $1\,°C$ at and south of the mooring position (Fig. 5c) was warmer than that in November 2019 (Fig. 5a). The isohalines were also aligned with the isotherms with salinities exceeding $34.2\,g\,kg^{-1}$ for the

water at $\geq 0\,°C$ at depth on the southern side. The northern side of the transect was below $-1.75\,°C$ as it was in November 2019.



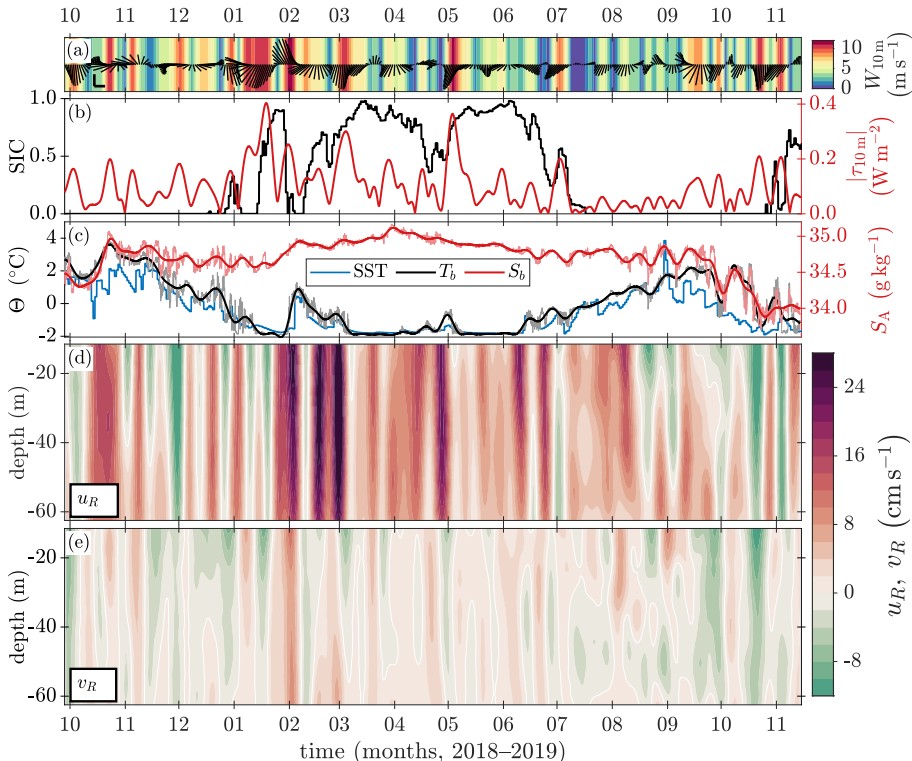

**Figure 6.** Time series over the mooring deployment period (29 September 2018 to 14 November 2019) of a) wind speed (colours) and wind velocity (quivers every 24 hours, an upward pointing vector is wind blowing northward, reference quivers are $5\,\mathrm{cm\,s^{-1}}$), b) sea ice concentration (SIC) (black, left axis) and wind stress (red, right axis), c) Conservative Temperature $\Theta\,(^\circ\mathrm{C})$ near the bottom (black), sea surface temperature (SST, $^\circ\mathrm{C}$) (blue), and Absolute Salinity $S_\mathrm{A}$ near the bottom (red, right axis), d) along-isobath ($-42^\circ$, directed southeastward) velocity component, $u_R$, and e) across-isobath ($+48^\circ$, northeastward) velocity component, $v_R$. Wind velocity, wind stress, mooring temperature and salinity, and current velocity are low-pass filtered with a cutoff-frequency corresponding to 10 days. $\Theta\,(^\circ\mathrm{C})$ and $S_\mathrm{A}$ are shown as both raw (pale, thin curves) and low-pass filtered (dark, thick curves). SST and SIC are daily values. Wind velocity, SST, and SIC were interpolated spatially onto the mooring position.

## 3.3 Seasonal variability in hydrography and current velocity from mooring records

The time series from the mooring shows near-bottom waters that were colder and less saline in October–November 2019 than in 2018 (Fig. 6c). This variability is in agreement with the substantial interannual variability uncovered in the hydrographic transects in the autumns of 2019–2021 (Fig. 5).


Early in the record, both the near-bottom temperature, $T_b$, and salinity, $S_b$, increased to values close to the pure AW properties by the middle of October (Fig. 6c). Temperature near the bottom was higher than the SST, and the increase in $T_b$ and $S_b$ coincided with a relatively strong and long-lasting surface-intensified current directed southeast along-isobath ($u_R$, Fig. 6d). Following the maxima in near-bottom temperature and salinity, the water column gradually cooled until mid-January 2019,





when both $T_b$ and SST nearly reached the freezing point (Fig. 6c). During these months, $u_R$ oscillated between $-5\,\mathrm{cm\,s^{-1}}$ and $10\,\mathrm{cm\,s^{-1}}$ at a two-week time scale with a negligible depth variability over the measured range (Fig. 6d). Temperature and salinity co-varied with $u_R$, with $T_b$ and $S_b$ lagging $u_R$ by 3.3 days and 4.8 days, respectively. Sea ice was first observed over the saddle in late December (Fig. 6b).

    From mid-January to mid-June 2019, there was a partial sea ice cover over the saddle (Fig. 6b), $T_b$ and SST were generally

at or close to the freezing point (Fig. 6c). $S_b$ increased, probably as a result of brine release from sea ice formation, reaching its highest value $35.12\,\mathrm{g\,kg^{-1}}$ by the end of March, gradually decreasing thereafter (Fig. 6c). The current was more unidirectional than during autumn, i.e. $u_R$ was generally positive (Fig. 6d) and varied in strength, typically between $5\,\mathrm{cm\,s^{-1}}$ and $10\,\mathrm{cm\,s^{-1}}$, episodically exceeding $20\,\mathrm{cm\,s^{-1}}$.

    Although the saddle was ice-covered and the water was generally near the freezing point during winter and spring, there

were several episodes when $T_b$ and SST increased, coinciding with a reduction of the SIC and an enhancement of $u_R$ (Fig. 6b–d). The first and largest increase in $T_b$ and SST occurred in early February 2019 after a sharp reduction of SIC at the end of January when the sea ice cover that had built up over two weeks during northerly winds disappeared in two days. The wind direction changed to southerly and $u_R$ changed to positive values shortly before the sea ice disappeared from the saddle (Fig. 6a and d). The surface-intensified $u_R$ increased in strength during and after the sea ice disappeared and reached a maximum of

$26\,\mathrm{cm\,s^{-1}}$ close to the surface (Fig. 6d). $T_b$ and SST increased to $0.9\,°\mathrm{C}$ and $0.4\,°\mathrm{C}$, respectively, shortly after and remained above the freezing point for more than a month (Fig. 6c). During this time, the wind direction turned and stayed northerly and the partial sea ice cover was reestablished over the saddle (Fig. 6a–b). $T_b$ and SST, however, remained above the freezing point and reached smaller maxima, and $u_R$ further increased in strength to $34\,\mathrm{cm\,s^{-1}}$ near the surface, the highest value measured during the whole sampling period (Fig. 6c–d).

The SIC started to decrease in June and the area was ice-free from mid-July (Fig. 6b). Most of the reduction of the sea ice cover occurred during two pulses of increased $u_R$ in the second half of June which preceded two pulses of increased $T_b$ and SST (Fig. 6d, c). After the sea ice cover had disappeared, $u_R$ became weaker, less unidirectional, and occasionally more depth-variable (Fig. 6d). From July through August, the water column steadily warmed. From September until the end of the measurement period in mid-November 2019, SST decreased toward the freezing point while $T_b$ was generally higher than SST.

$S_b$ generally co-varied with $T_b$ during this autumn and overall decreased (Fig. 6c). By late October 2019, some sea ice had already drifted over the saddle and SST reached the freezing point (Fig. 6b, c), while $T_b$ remained above the freezing point until the mooring was recovered in mid-November (Fig. 6c).

### 3.4   Mesoscale and tidal variability

Wavelet analysis of current components $u_r$ and $v_r$ (along $-28\,°$ and $+62\,°$, respectively); shows that semidiurnal tidal currents

dominate the variability throughout the year with a clear spring–neap cycle (Fig. 7). The rotary spectral estimates (Fig. 7f–g) indicate that the semidiurnal tidal current is anticyclonic. The semidiurnal band is also energized in the near-bottom temperature $T_b$ record during periods when the temperature is above the freezing point (Fig. 7d–e).





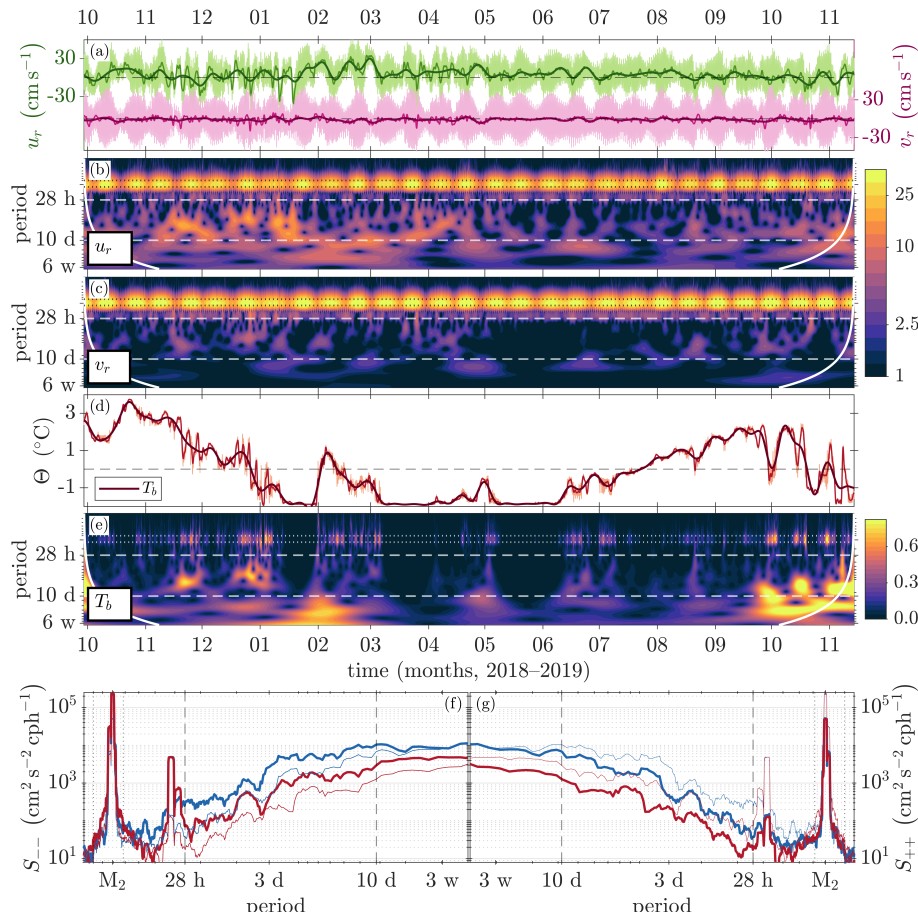

**Figure 7.** a) Time series over the mooring deployment period (29 September 2018 to 14 November 2019) of depth-averaged velocity component along $-28°$, $u_r$ (green, left axis) and along $+62°$, $v_r$ (pink, right axis), as measured (pale, thin curves), 28 hour low-pass filtered (darker, thicker curves), and 10 day-low-pass filtered (darkest, thickest curves). b–c) Wavelet transform of $u_r$ (b) and $v_r$ (c) in logarithmic scale. d) Time series of near-bottom temperature $T_b$ with the same filtering as in (a). e) Wavelet transform of $T_b$ in linear scale. f–g) Rotary spectra of the depth-averaged current velocity for winter (October to March, blue curves) and summer (April to September, red curves). In (f), the anticyclonic rotary spectra are shown as thick curves and the cyclonic spectra as thin curves. In (g) the cyclonic rotary spectra are shown as thick curves and the anticyclonic spectra as thin curves. White curves in (b,c,e) show the area affected by edge effects. The cutoff frequencies for the semidiurnal tidal band (dotted lines) and the weather band (dashed lines) are shown as horizontal lines in (b,c,e) and vertical lines in (f,g).

Considerable variability is observed in the low-frequency band corresponding to periods between 28 hours and 10 days (between dashed lines in Fig. 7b, 7c, 7e), particularly during the autumn months (November 2018 into January 2019). This variability is also seen in the time series of 28-hour low-pass filtered $u_r$ (Fig. 7a) and $T_b$ (Fig. 7d). In the beginning (October




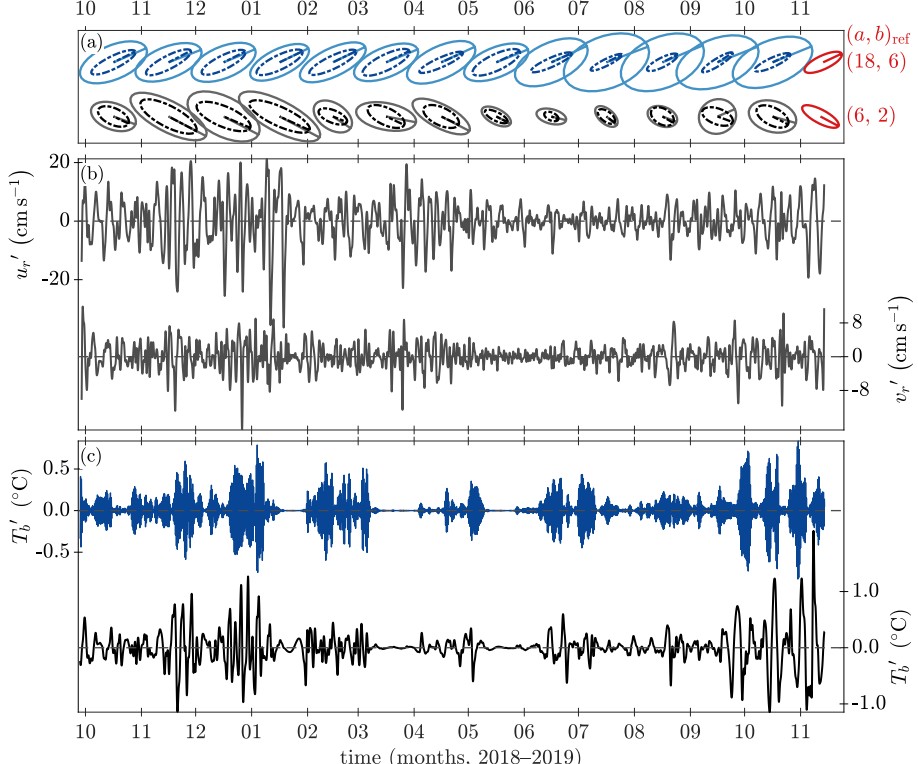

**Figure 8.** Time series over the mooring deployment period (29 September 2018 to 14 November 2019) of a) semidiurnal variance ellipses (blue) and weather-band variance ellipses (black) for the current at $14.5\,\mathrm{m}$ (solid curve, pale colour) and at $59.9\,\mathrm{m}$ (dashed curve, dark colour), b) weather-band depth-averaged current velocity anomalies along $-28\,^\circ$ ($u_r{}'$, left axis) and along $+62\,^\circ$ ($v_r{}'$, right axis), c) temperature anomalies in the semidiurnal band (blue, left axis) and the weather band (black, right axis). In (a), the weather-band ellipses are scaled up three times relative to the semidiurnal ellipses; reference ellipses (red) with semi-major and semi-minor axes $(a, b)_{\mathrm{ref}}$ given for both bands have orientations $\theta = +25\,^\circ$ and $\theta = -28\,^\circ$ for semidiurnal and weather band, respectively.

2018) and toward the end of the record (October–November 2019), $T_b$ contains elevated variance at the time scale on the order 10 days; however, the edge effects of the wavelet analysis are significant.

The anticyclonic velocity component is generally more energetic than its cyclonic counterpart (Fig. 7f–g). Specifically, the semidiurnal band has almost ten times more variance in the anticyclonic component than the cyclonic one, and is approximately
two orders of magnitude more energetic than the diurnal band. At lower frequencies, the currents are anticyclonic elliptically polarized during summer. However, during winter, the currents are nearly linearly polarized.

The temporal evolution of variability in the semidiurnal tidal band and the weather band is shown in Fig. 8. The variance ellipses of the semidiurnal current velocity are oriented along $+25\,^\circ$, i.e., near east-northeast. The ellipses are dominated by the $M_2$ tidal constituent (Fig. 7f–g), which has an anticyclonic rotation. From October 2018 to June 2019, the semidiur-
nal current was predominantly barotropic (Fig. 8a, blue ellipses), showing little variation except for the spring–neap cycle





(Fig. 7a–c) due to interaction between the $M_2$ and $S_2$ tidal constituents. During these months, the ellipse amplitude was $\kappa = (23.6 \pm 1.1)\,\mathrm{cm\,s^{-1}}$ near the surface and $\kappa = (16.2 \pm 0.6)\,\mathrm{cm\,s^{-1}}$ near the bottom. However, from June to October 2019, the semidiurnal current was considerably more baroclinic, with larger $\kappa = (30.2 \pm 2.0)\,\mathrm{cm\,s^{-1}}$ near the surface, and smaller $\kappa = (12.1 \pm 1.3)\,\mathrm{cm\,s^{-1}}$ at the bottom. During this period, the semidiurnal current in the upper water column became more

circularly polarized, while in the lower water column, it became more rectilinear.

The semidiurnal variability in $T_b$ (Fig. 8c, represented by the dark-blue curve) shows anomalies up to $0.8\,^\circ\mathrm{C}$, but typically within the range of $0.2\,^\circ\mathrm{C}$ to $0.6\,^\circ\mathrm{C}$. Although the spring–neap cycle is visible, other factors also affect the amplitude of the temperature anomalies.

The principal axis of current variability in the frequency band corresponding to periods between 28 hours and 10 days

is approximately aligned with the topography (along the WNW–ESE direction) (Fig. 8a, black and gray ellipses). Between October 2018 and January 2019, the monthly variance ellipses for the upper and lower water columns are oriented between $-20\,^\circ$ and $-30\,^\circ$ and have amplitudes of $4\,\mathrm{cm\,s^{-1}}$ to $10\,\mathrm{cm\,s^{-1}}$. The ellipse amplitudes decrease until July and gradually increase toward the end of the record. Temperature anomalies in this frequency band were typically within $\pm 0.5\,^\circ\mathrm{C}$, increasing twofold in November–December 2018 and October–November 2019, reaching a maximum of $2\,^\circ\mathrm{C}$ in early November 2019.

### 3.5 Impact of wind forcing on the across-saddle current

The strength and direction of the overflow current are significantly influenced by large-scale winds (Fig. 9). Regression analysis shows that the depth-averaged current anomalies along $-28\,^\circ$ in the frequency band from 28 hours to 10 days ($u_r'$) depend on the Ekman transport along the same axis. Geostrophic adjustment to Ekman transport during north-northeasterly winds typically opposes the east-southeastward cross-saddle current (see Fig. 9c), and in the case of anomalously strong winds, even

reverses it (see Fig. 9b). Conversely, weaker and/or southerly winds tend to enhance the eastward flow into the Barents Sea (Fig. 9d).

Anomalously strong north-northeasterly winds, causing Ekman transport values below the 10th percentile ($U_{E_r} = (-1.84 \pm 0.53)\,\mathrm{m^2\,s^{-1}}$ for daily averages), are associated with the strongest current reversal events ($u_r' = (-5.1 \pm 8.8)\,\mathrm{cm\,s^{-1}}$ for daily averages, with current flowing towards the WNW along topography) (Fig. 9b). The mean current velocity is then $3.1\,\mathrm{cm\,s^{-1}}$, directed

west-southwestwards, but there is a large spread.

On the other hand, winds with a southerly component, causing Ekman transport values above the 90th percentile ($U_{E_r} = (0.79 \pm 0.35)\,\mathrm{m^2\,s^{-1}}$ for daily averages), are associated with the strongest east-southeastward current events ($u_r' = (4.4 \pm 6.3)\,\mathrm{cm\,s^{-1}}$ for daily averages) (Fig. 9d). This suggests that the normal east-southeastward current is accelerated by geostrophic adjustment to Ekman transport set up by winds with a southerly component. The average mean current in this case is $14.1\,\mathrm{cm\,s^{-1}}$ towards

the east-southeast.

### 3.6 Eastward transport of positive temperature anomalies

The near-bottom temperature exhibits variability in frequency bands that overlap with those of the current (Fig. 7). We analyse anomalies associated with tides, weather, and mesoscale activity obtained by band-passed filtering of temperature and velocity



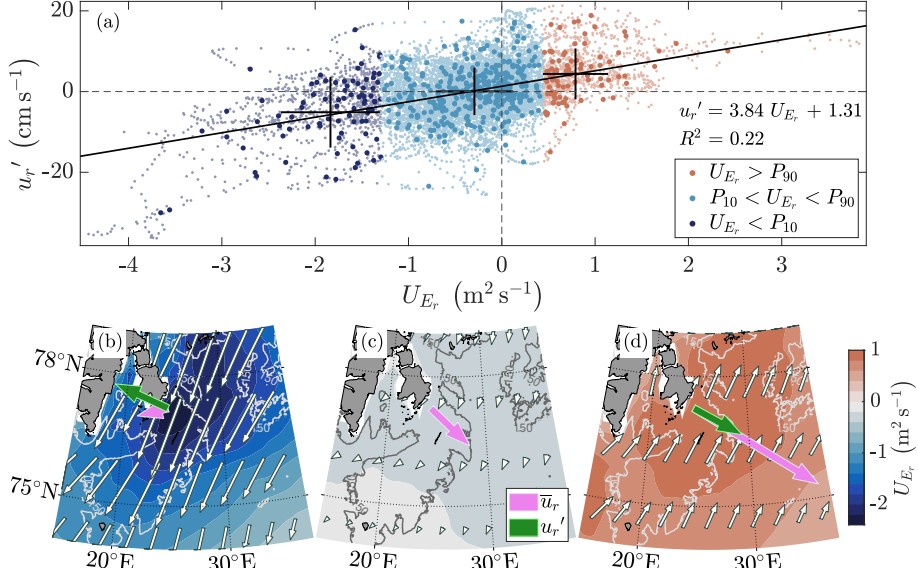

**Figure 9.** a) Ekman transport $U_{E_r}$ along $-28\,°$ versus the weather-band (28 h – 10 d) depth-averaged current speed anomalies along $-28\,°$ $u_r{'}$. Colours indicate the $U_{E_r}$ composites: below its 10th percentile (dark blue), corresponding to strong north-northeasterly winds, within its 10th and 90th percentiles (light blue), corresponding to weak north-northeasterly winds, and above its 90th percentile (orange), corresponding to modest south-southwesterly winds. The small, transparent dots are hourly values, while the larger dots are daily averages. The current is lagged by 7 hours relative to the wind before calculating $U_{E_r}$ and averaging in time. The error bars show the mean and standard deviations of the daily averages within each composite. Maps of composite averages of $U_{E_r}$ (colour shading), wind stress (white quivers), mean current $\overline{u}_r$ (pink quivers), and current anomalies $u_r{'}$ (green quivers) for Ekman transport b) below its 10th percentile, c) within its 10th and 90th percentiles, and d) above its 90th percentile.

records (Sect. 2.2). The velocity components are rotated along the principal axis, $-28\,°$, of the weather-band variability. By
analysing temperature anomalies, $T'$ in the velocity anomaly space $(u', v')$, we find that positive temperature anomalies are generally located in the southeast sector of the velocity anomalies, while negative temperature anomalies are in the northwest sector, resulting in net temperature transport across the saddle in the identified bands of variability (Fig. 10a– 10c). For the semidiurnal (Fig. 10a) and also diurnal tidal bands (not shown), the largest positive temperature anomalies tend to be oriented in the southeast sector, nearly perpendicular to the direction of maximum variance (major axes of the ellipses). For the fre-
quency bands corresponding to the weather and mesoscale activity (Fig. 10b and Fig. 10c, respectively), the largest temperature anomalies and the largest variance tend to align approximately parallel to each other, both located in the southeast sector.

The eddy temperature flux $\overline{u_r{'}T'}$ along $-28\,°$, (Fig. 10d–10f, solid curves) in all analysed time scales is generally positive through the year, with values typically between $1\,\mathrm{K\,cm\,s^{-1}}$ and $2\,\mathrm{K\,cm\,s^{-1}}$. The largest contribution to the eddy temperature flux from the semidiurnal tide (Fig. 10d) occurs during October-December in both years, with values on the order of $1\,\mathrm{K\,cm\,s^{-1}}$
directed towards the southeast. Comparable fluxes are also calculated between February and July when warm and saline water intermittently appears near the saddle. In the weather band, $\overline{u_r{'}T'}$ exhibits large values ranging from $1\,\mathrm{K\,cm\,s^{-1}}$ to $3\,\mathrm{K\,cm\,s^{-1}}$




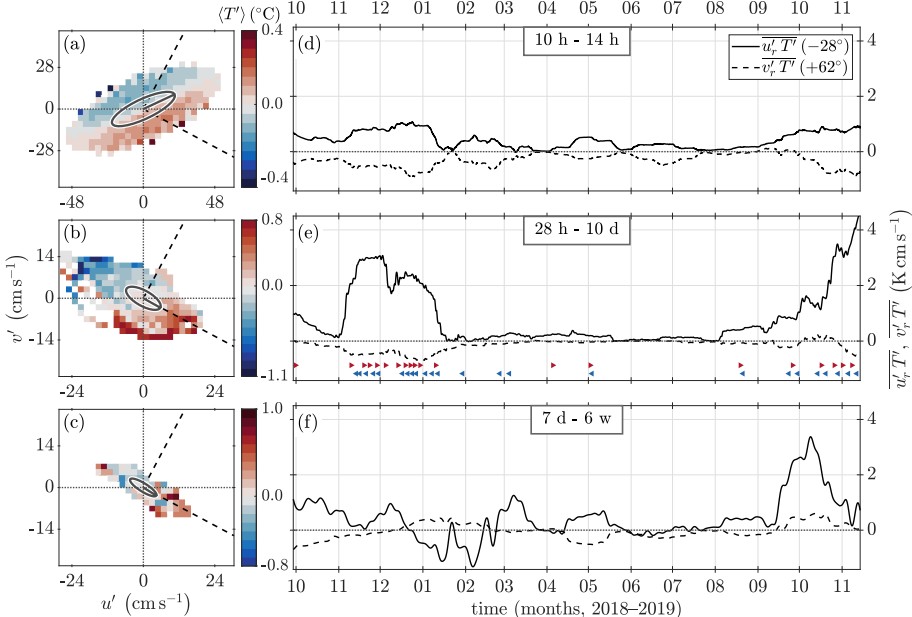

**Figure 10.** a, b, c) Temperature anomaly (colours) averaged in bins of current velocity anomalies analysed in three frequency bands. The velocity anomalies are layer-averaged over the bottom half of the water column, and their variance ellipses are overlaid. The frequency bands correspond to the semidiurnal tidal band (a, d), weather band (28 hours to 10 days; b, e), and lower-frequency band (seven days to six weeks; c, f). d, e, f) Time series of eddy temperature flux along $\overline{u_r{}'T'}$ ($-28\,^\circ$, solid curves) and across $\overline{v_r{}'T'}$ ($+62\,^\circ$, dashed curves) the principal axis of the weather-band variability shown in (b). The rotated coordinate system is indicated with dashed lines in the left column. Temporal averaging is done by running mean over 30 days. Note that the $u'$ and $v'$ ranges in the left column are twice as high for the semidiurnal band as for the weather and lower-frequency bands. In the weather band time series (e), the times of peaks chosen for ensemble averaging are marked with red triangles pointing right for positive temperature and along-isobath current anomalies, and blue triangles pointing left for negative anomalies.

from November 2018 to January 2019, and then in autumn 2019, reaching $5\,\mathrm{K\,cm\,s^{-1}}$ by the end of the measurement period (Fig. 10e). Between mid-January and August 2019, however, the eddy temperature flux is negligible. Lower frequency oscillations on timescales between one and six weeks exhibit sporadic episodes of large positive eddy temperature flux values during

autumn and winter, reaching $3\,\mathrm{K\,cm\,s^{-1}}$ in October 2019, and reversals with negative values during January and February 2019. (Fig. 10f).

    The large positive temperature anomalies in the weather band correspond to velocity anomalies in the SE sector, while negative anomalies correspond to those in the NW sector (Fig. 10a–10c), both resulting in a positive eddy temperature flux toward east-southeast (Fig. 10d–10f, solid curves). To further investigate the driving force behind these anomalies, we analysed

ensembles of strong eddy temperature flux events in the weather band and examined the associated anomalies in observed temperature, current velocity, and wind forcing. In total 20 events with peaks $(u_r{}', T') > 0$ and 24 events with peaks $(u_r{}', T') < 0$ were detected, both resulting in positive temperature fluxes. Most of these events occur during October, November, and



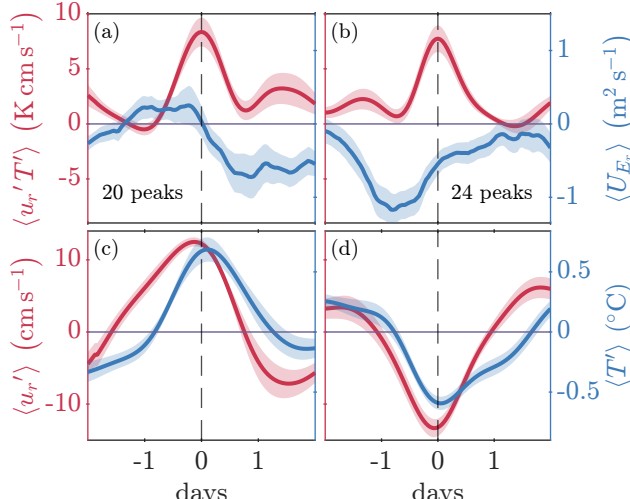

**Figure 11.** Ensemble-averaged time series for events with strong positive eddy temperature flux $\langle u_r' T' \rangle$ (red) along the direction of highest variance ($-28\,^\circ$) in the weather band (28 hours to 10 days) for the cases when $(u_r', T') > 0$, i.e., positive temperature anomalies are carried ESE by the current anomaly along isobath (a and c), and the cases when $(u_r', T') < 0$, i.e., negative temperature anomalies are carried WNW by the current anomaly along isobath (b and d). The ensemble time series of Ekman transport component along $-28\,^\circ$, $\langle U_{E_r} \rangle$, are shown in blue (a-b), of current anomalies $\langle u_r' \rangle$ in red (c-d), and of temperature anomalies $\langle T' \rangle$ in blue (c-d). The current anomalies are bin-averaged over the bottom half of the water column. The time axes (days) are centered at the peak of the eddy temperature flux. The standard error ($\mathrm{SE} = \mathrm{std}(x)/\sqrt{n}$, where $n$ is the number of ensembles) is shown with shading.

December (red and blue triangles in Fig. 10e) when there is no sea ice in the area. The time series of 4-day duration centered at the selected time of each event are extracted. The ensemble-averaged records are denoted using angle brackets, $\langle . \rangle$ (Fig. 11).

The ensemble of events that satisfy the first condition is associated with southerly wind stress, resulting in an Ekman transport component that forces a positive current anomaly carrying positive temperature anomalies toward the east-southeast (Fig. 11a and 11c). The peak eddy temperature flux of $(8.4 \pm 1.3)\,\mathrm{K\,cm\,s^{-1}}$ tends to occur one day after the maximum along-isobath Ekman transport. The temperature anomaly turns positive approximately half a day after the current anomaly, and their peaks are usually six hours apart.

The second case (Fig. 11b and 11d; blue triangles in Fig. 10e) corresponds to events in which the wind from the north-northeast increases in strength, leading to an enhancement of the typical Ekman transport opposing the current and hence a negative current anomaly carrying negative temperature anomalies toward west-northwest. The peak of eddy temperature flux reaches $(7.8 \pm 1.2)\,\mathrm{K\,cm\,s^{-1}}$, approximately one day after the along-isobath Ekman transport component has reached its largest negative value. The temperature anomaly turns negative six hours after the current anomaly, and their peaks are two 295   hours apart.



## 4    Discussion

### 4.1    Interannual vs seasonal variability – was 2018–2019 a typical year?

Several findings in the time series at the mooring position (Fig. 6) and the late-autumn hydrographic transects on the saddle (Fig. 5) warrant a discussion about the seasonal and interannual variability in the area and whether the 2018–2019 was a typical

year.

A clear difference between the autumns of 2018 and 2019 in terms of hydrography and the onset of the sea ice period was evident. The recent hydrographic transects done in November 2019, October 2020, and November 2021 (Fig. 5) indicate that interannual variability may be a typical feature in the area. In the autumn of 2018, the water column above the saddle was warm, remained ice-free and was only partially covered with ice from December. In contrast, in the autumn of 2019, the sea

ice drifted to the saddle in late October and the water had cooled close to the freezing point in November. This difference could indicate that the position of the ice cover in the Barents Sea during the autumn is relevant for the hydrographic conditions over the saddle at this time of the year. According to Kohlbach et al. (2023), the marginal ice zone had retreated beyond the shelf break into the Nansen Basin in August 2018, while in August 2019 large parts of the northwestern Barents Sea shelf were still covered by sea ice, resulting in lower temperatures and a fresher surface layer (Kohlbach et al., 2023). These differences in

the large-scale conditions at the end of the melting seasons of 2018 and 2019 must also have affected the following winter conditions of 2018–2019 and 2019–2020 observed at our mooring site.

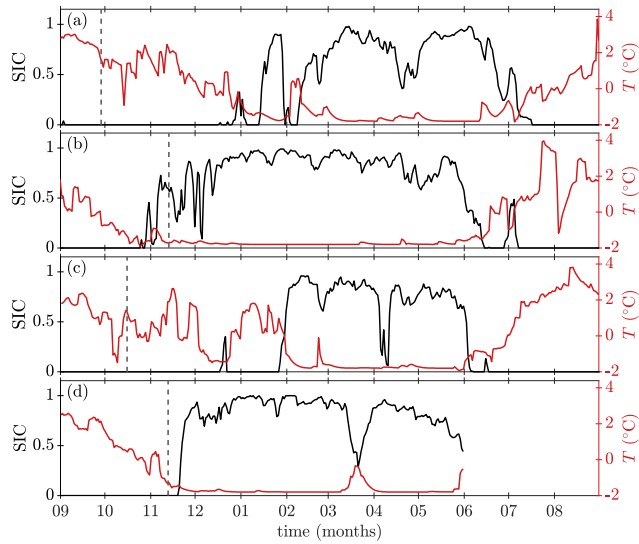

**Figure 12.** Sea ice concentration (black, left vertical axis) and sea surface temperature (red, right vertical axis) from September to September in a) 2018–2019, b) 2019–2020, c) 2020–2021, and d) 2021–2022. Dashed lines mark times of a) mooring deployment, b) mooring recovery and hydrographic transect in November 2019, c) hydrographic transect in October 2020, and d) hydrographic transect in November 2021.



During the weeks before the transect in November 2019 (Fig. 5a) was taken, winds were predominantly northeasterly (Fig. 6a) with relatively strong current reversals towards west and northwest (Fig. 6d–e), presumably importing sea ice and contributing to lowering the sea surface temperature to the freezing point above the saddle. Comparing the large-scale sea ice conditions in November of 2019 and 2021, the $15\,\%$ sea ice edge extended past Hopen and almost reached the southern tip of Spitsbergen in 2019, while in 2021, it only extended south to Edgeøya and did not cover the Olga Basin (not shown). The higher temperature maximum and the alignment of the isohalines and isotherms in the transect from November 2021 (Fig. 5c) suggest that there had been an inflow of water of Atlantic origin onto the saddle during autumn and that the water column had not yet cooled. The SST in the region was substantially higher in 2021 compared to 2019, with the largest difference of $3\,°C$ in Storfjordrenna (not shown), and $1\,°C$ to $2\,°C$ at the mooring position, so that the onset of freezing was delayed by one month in 2021 relative to 2019 (Fig. 12b, 12d).

The November transects (Fig. 5a and 5c) were both cooler and less saline than the October transect (Fig. 5b). This can partly be explained by the time difference as October is at the start of the autumn and the water column has not yet cooled down at that time. However, interannual variability may play a larger role. During autumn 2020, SST steadily decreased until rapidly increasing to $1.4\,°C$ during the week before the transect in mid-October (Fig. 12c). It then decreased to $0\,°C$ before increasing to $2.6\,°C$ in mid-November. Hence, autumn 2020 had warmer and longer-lasting overflows of water of Atlantic origin on the saddle and a later onset of sea ice than the other years.

Autumn 2018 was comparable to autumn 2020 in terms of SST evolution at the mooring position (Fig. 12a and 12c). Inflow of warm water from Storfjordrenna maintained high SST above the saddle through November in both 2018 and 2020, and delayed the onset of sea ice in the subsequent winters.

In the context of the long-term changes in SST and SIC at the study site, the autumn 2018 to autumn 2019 (mooring year) was an average year in the recent two decades (not shown), and the difference between 2018 and 2019 was not remarkably large. However, the mean SST in 2018–2019 was higher than all years between 1981 and 2006. In 2018–2019, there were 192 days of sea ice and 81 days of SIC above 0.8. For reference, in 2015–2016, the mean SST above the saddle was abnormally high, and there were only 106 days of any sea ice and no days SIC above 0.8 (not shown). This was the same year when Storfjordrenna, along with the Barents Sea in general, experienced a record-low sea ice cover and a warm and saline surface layer (Vivier et al., 2023), and a record-warm and -long-lasting marine heatwave (Bayoumy et al., 2022a).

Our results show that, in addition to the normal seasonal cycle, there were several episodes of warm water intrusions captured by the mooring, which affected the sea ice cover (Fig. 6b–c). The inflow of Atlantic-origin waters and heat transport onto the saddle are important for the onset and length of typical winter conditions. Stronger current velocities directed across-saddle into the Barents Sea ($u_R$) generally preceded the local temperature maxima associated with the warm water intrusions (Fig. 6c and 6d).

The sea ice covering the saddle during the second half of January 2019 occurred before the water column had completely cooled to the freezing point (Fig. 6b, c). The SIC increased while there were strong westerly to northerly winds (Fig. 6a), and the evolution of the large-scale sea ice cover revealed that sea ice was imported from the Olga Basin toward the saddle (not shown). When the wind ceased and then picked up with a southerly component, the sea ice edge drifted northeastwards away





from the saddle. Concurrently, the across-saddle current became stronger as a result of geostrophic adjustment to the wind-driven Ekman transport from Edgeøya to Hopen. After the sea ice edge had moved north of the saddle, the current strength further increased, reaching a depth-averaged value of $23\,\mathrm{cm\,s^{-1}}$, eventually driving warmer waters from Storfjordrenna to the
saddle. This kind of deep warm water overflow response to wind forcing has also been observed on the sill in the inner part of Hornsund in Svalbard (Arntsen et al., 2019).

A second period characterized by strong across-saddle current and increased temperature occurred from mid-February to March. In contrast to the previous current and temperature maxima, this increase cannot be explained by local wind. Throughout this period, northerly winds brought sea ice from the Barents Sea interior. There was no strong temperature increase
associated with this current maximum, however, the SST and the near-bottom temperature remained above the freezing point into March and increased slightly during the highest current maximum of $30\,\mathrm{cm\,s^{-1}}$ at the end of February. This increase is most likely caused by upstream conditions. One possible cause could be an acceleration of the WSC and the branch going into Storfjordrenna which can force the current to follow shallower isobaths to conserve potential vorticity, as demonstrated by a barotropic shelf circulation model (Nilsen et al., 2016). The small temperature increase during this period was most likely due
to mixing of the warm water en-route to the saddle. When the current subsided in early March, the temperature on the saddle returned to the freezing point, and the sea ice concentration increased to $90\,\%$.

Similarly, the instances with increased temperatures in April are likely forced by the upstream conditions. The effect of the current pulses was smaller as the water transported to the saddle were cooler than in February, especially near the surface. Changes in the local sea ice concentration seemed to be mostly affected by the local wind, not the current and near-bottom
temperature.

To summarise, the intrusions of warm water during autumn can delay the onset of winter conditions on the saddle and inhibit local sea ice growth or melt sea ice that has been imported from northeast, as observed in the autumn of 2018 (Fig. 6b–c and 5b). Further, such warm water intrusions, whether they are forced by local winds or by upstream conditions, frequently reduce the existing partial ice cover above the saddle during winter and spring (Fig. 12, earlier years not shown). This explains why
November 2019 and November 2021 were substantially different (Fig. 5a and 5c) and that October 2020 was much warmer not only because the section was taken one month earlier and cooled less, but also because of the more persistent warm water overflow that year.

### 4.2 Drivers of the mean exchange between Storfjordrenna and Barents Sea

The observed current (10-day low-passed) tends to flow towards the southeast (Fig. 6c). This current direction is opposed by
the geostrophic adjustment to the Ekman transport induced by the typical large-scale winds from the northeast. The strength and direction of the current are affected by the wind stress, and in cases with strong winds, either reversed to northwestward flow or amplified in its normal southeastward direction (Fig. 9). Hence, the local wind is not the main driver of the background current on the saddle.

The forcing of the slowly varying current is more complex. A low but insignificant correlation between spatial differences
in sea level and mooring current velocity was found by analysing absolute dynamic topography and the associated geostrophic





currents (not shown). This means either that the sea level slope locally is not the main driver of the current or that the data coverage and quality of the absolute dynamic topography measurements are not sufficiently good to show a clear relation. The quality of such measurements can be reduced due to seasonal ice coverage and the timing of the satellite passages, and the product quality north of the Arctic circle is not thoroughly tested (Oziel et al., 2020).

The current across the saddle may, on the other hand, be dependent on the strength of the current upstream in Storfjordrenna and on the slope west of Spitsbergenbanken (Sect. 4.1). Brown et al. (2023) showed that further upstream, along the Norwegian coast, the Norwegian Atlantic Current gets accelerated by low-pressure systems through onshore Ekman transport and set-up along the coast. However, it is unclear whether the passing of low-pressure systems would accelerate the WSC since West-Spitsbergen has a shorter coast than the Norwegian coast (Brown et al., 2023). In addition, Wickström et al. (2020) showed

that there is high variability in the track of the passing low-pressure systems toward and past Svalbard, and therefore there is also highly variable local wind forcing west of Storfjordrenna and Svalbard. Peaks in the current on the saddle seem at times to be related to a strong WSC quantified using the slope of absolute dynamic topography between Spitsbergenbanken and Storfjordrenna (not shown) generating a barotropic contribution to the saddle current strength. Similar to the response of the WSC on the West Spitsbergen Shelf to anomalous wind stress curl (Nilsen et al., 2016), the branch of the WSC which flows

in Storfjordrenna can be displaced onto shallower isobaths. However, a more detailed analysis is needed to conclude on the effects of upstream forcing.

Another possible driver for the observed mean current is the tidal residual flow on Spitsbergenbanken. In shallow areas with strong tidal currents and strong non-linear bottom friction, residual currents may influence the circulation pattern (Harms, 1992). Tidally generated residual currents around Bjørnøya were reported from buoys (Vinje et al., 1989) and from lab-

oratory experiments and observations from drifters (McClimans and Nilsen, 1993). Several numerical model studies have shown that tidal rectification occurs in the region (Harms, 1992; Gjevik et al., 1994; Kowalik and Proshutinsky, 1995). Using higher-resolution models which better resolved nonlinear terms, strong anticyclonic tidal residual current was reported around Bjørnøya and on Spitsbergenbanken (reaching $15\,\mathrm{cm\,s^{-1}}$) and around Hopen (Kowalik and Marchenko, 2023). Simulations have reproduced the anticyclonic drift of buoys around Hopen, which is driven by strong tidal currents caused by interaction

between the semidiurnal tide and the island of Hopen (Marchenko and Kowalik, 2023).

### 4.3   Tidal-, weather- and low-frequency band variability and their impact on the eastward heat transport into the Olga Basin

Both the semidiurnal and the diurnal tidal currents contribute to the transport of heat across the saddle on Hopenbanken. The semidiurnal contribution is significantly larger. The semi-major axis of the semidiurnal currents is oriented along $20°$, i.e.

roughly east-northeast, which is roughly perpendicular to the eddy temperature flux in the semidiurnal band (Fig. 10). The direction of the eddy temperature flux is mainly in the southeast sector for all the frequency bands. This is likely due to the position of the temperature gradient on the saddle and the guiding of the current anomalies by topography.

The oscillations on timescales between one and six weeks result in sporadic moderate eddy temperature fluxes in the southeast sector, comparable to the semidiurnal band in magnitude, but more variable in direction. The largest contribution towards





east-southeast was during autumn 2019 (Fig. 10f), when the wavelet transform of $u_r$ (Fig. 7b), and especially of $T_b$ (Fig. 7e), had power in the low-frequency area, implying that the varying current brought even larger temperature variations. These concurrent fluctuations added up to a considerable eddy temperature flux in this period. We hypothesise that the high eddy temperature flux this autumn may be caused by AW upstream of the saddle being lifted onto shallower isobaths due to either regional Ekman pumping or an acceleration of the upstream current system, resulting in higher temperatures on one side of the front on the saddle. Locally on the saddle, this was observed as low-frequency oscillations with particularly high-temperature variations causing a large eddy temperature flux into the Barents Sea.

Integrated over the measurement period, the weather band eddy temperature flux was approximately twice that of the low-frequency and the semidiurnal bands, both of which had comparable magnitudes. During the period with a relatively high sea ice coverage, from February through July, the semidiurnal tide leads twice as much heat across the saddle as the weather band and 1.5 times as much as the band associated with mesoscale activity.

Eddy temperature fluxes calculated in Section 3.6 contain both their divergent and rotational components. Only the divergent part plays a part in the actual transport of heat across the saddle (Marshall and Shutts, 1981; Guo et al., 2014). Using our data set we cannot separate the contribution of the divergent or rotational eddy temperature fluxes, hence, the values must be taken as an upper bound.

### 4.4 Consequences of the long-term changes in the hydrographic environment in Storfjordrenna for the heat transport into the Barents Sea

Over time, the AW presence in Storfjordrenna upstream of the study site has increased. Storfjordrenna is among the locations in the Barents Sea that have experienced the highest increase in sea surface temperature over the recent two decades (Barton et al., 2018; Bayoumy et al., 2022b). The climatological hydrographic maps and transects in Storfjordrenna (Section 3.1) show that also the water column has become warmer since 2000 compared to previous years (Fig. 3), and that Atlantic-origin water has shoaled and now reaches further eastwards towards the saddle during summer (Fig. 4). The shoaling of AW has also been observed in the Eurasian Arctic (Polyakov et al., 2017), along west-Spitsbergen, and in the fjords along West-Spitsbergen (Tverberg et al., 2019; Skogseth et al., 2020; Strzelewicz et al., 2022). In addition, AW advected into the fjords in West-Spitsbergen have warmed in the recent two decades (Bloshkina et al., 2021; Pavlov et al., 2013; Tverberg et al., 2019; Skogseth et al., 2020). As the AW circulates over shallower isobaths, the physical processes taking place on the saddle between Storfjordrenna and the Olga Basin may more easily transport heat eastward in the future. Anomalous wind curl can affect the circulation depth of AW in Storfjordrenna and therefore lead AW or Atlantic-origin waters closer to saddle between Edgeøya and Hopen (Sect. 4.2).

Several factors determine the fate of the AW crossing the saddle. The circulation pattern on and east of the saddle affects whether AW is transported north or south of the 200 m deep sill separating the Olga Basin from Hopendjupet and Bjørnøyrenna (Fig. 2), i.e. whether to the Arctic or the Atlantic side of the Polar Front. The time-averaged flow observed by the mooring was directed towards the southeast and most of the non-tidal variability contributing to heat exchange occurred in the axis between west-northwest and east-southeast (Sections 3.4, 3.5, 3.6). However, the bathymetry on the saddle is complex, and the





observations do not confirm that the current is strongly constrained by the topography. Without more spatially diverse current observations, we cannot conclude on the path of the AW, or the fraction of AW and heat that could contribute to the heat budget on the northern side of the Polar Front. Potentially, the AW transport through the channel between Edgeøya and Hopen can have consequences for the Olga Basin which still has a seasonal ice cover and a cold, thick layer of Arctic Water (e.g., Lind and Ingvaldsen, 2012).

Given the possibility that AW from Storfjordrenna may be transported into the Arctic side of the Polar Front, the density within the Olga Basin is an important factor in determining the consequences of the transport of Atlantic-origin waters for the local conditions. Denser waters on the eastern side would cause Atlantic-origin waters to penetrate at shallower depths and impact the sea ice cover during winter and spring and generally depositing more heat into the upper water column with implications for the sea ice growth in the following season. Less dense waters on the eastern side could on the other hand cause the overflow waters to subduct. If pure AW or water with properties closer to AW than what was observed in this study flows over the saddle, its density can exceed that of the local water in the Olga Basin (Arctic Water, East Spitsbergen Water) and thereby contribute to the deeper stratification.

Atlantic-origin waters reaching the saddle between Storfjordrenna and the Barents Sea will affect the properties of the East Spitsbergen Current flowing into Storfjorden. More AW mixed into the East Spitsbergen Current on the saddle will have consequences for the Coastal Current in Storfjorden and along the west coast of Spitsbergen and for the exchange between the shelf and the fjords. More AW in the East Spitsbergen Current will also impact the dense water production in Storfjorden and the consequent deep water export to Fram Strait, as the initial salinity of the source water preconditions the density of the overflow (Schauer, 1995; Skogseth et al., 2004; Vivier et al., 2023).

## 5 Conclusions

We have used available historical hydrographic profiles to study changes in the hydrographic environment in Storfjordrenna, the shallow banks around, and in the Olga Basin north of the Polar Front in the recent two decades compared to 1930–2000. In addition, we showed recent late autumn hydrographic transects and a year-long time series of near-bottom hydrography and water column current velocity on the saddle separating Storfjordrenna from the Olga Basin.

The mooring observations show that AW and Atlantic-origin waters in Storfjordrenna can cross the saddle and enter the Arctic domain of the northwestern Barents Sea. The across-saddle current is mediated by wind forcing. Southerly winds over the saddle cause stronger currents into the Barents Sea and are associated with events of strong eddy temperature flux. The strong semidiurnal tidal currents over the saddle also contribute to heat transport from Storfjordrenna into the Barents Sea. The area is characterised by large seasonal and interannual variability in hydrography and sea ice cover and frequent intrusions of warm water from Storfjordrenna. Both local wind stress and upstream conditions can force these intrusions, and they are important for the onset and length of typical winter conditions in the area.

The two recent decades were warmer and more influenced by AW in the Atlantic sector of the northwestern Barents Sea. This increase in heat in the Atlantic sector, potential shoaling of AW, and the ongoing changes in large-scale wind patterns





would indicate that the channel separating Storfjordrenna from the Barents Sea may emerge as an important pathway of AW and heat into the Arctic sector of the Barents Sea in the future.

*Data availability.* The mooring data can temporarily be retrieved from https://figshare.com/s/e4f2c67e8df9f68d306b. This data set will
be published in the Norwegian Marine Data Centre. Hydrographic data (Skogseth et al., 2019) used for the climatological sections can be retrieved from https://data.npolar.no/dataset/39d9f0f9-af12-420c-a879-10990df2e22d. ERA5 reanalyses (Hersbach et al., 2020) are available from https://cds.climate.copernicus.eu/cdsapp#!/dataset/reanalysis-era5-single-levels. The Global Ocean OSTIA Sea Surface Temperature and Sea Ice Reprocessed product (OSTIA, 2023; Good et al., 2020) is available from https://data.marine.copernicus.eu/product/SST_GLO_SST_L4_REP_OBSERVATIONS_010_011/description.

**Appendix A:  Methodology for the climatological hydrography maps and sections**

The historical hydrographic data have been optimally interpolated onto horizontal and vertical climatological sections to compare the two recent decades to earlier decades. The interpolation was done using the kriging algorithm in Golden Software Surfer 12 through the MATLAB function surfergriddata.m (point kriging with no drift, linear variogram model with unit slope and anisotropy). The spatial resolution of bin averages before kriging is described in section 4.2–4.3 of Skogseth et al. (2020).
Due to temporally and spatially sparse data coverage, only summer (July–October mean) sections were possible to create. For the period 1930–2000, if a bin had data from fewer than three years, that bin was retained without data to ensure an unbiased representation. For the period 2000–2019, this minimum threshold was lowered to one year to ensure enough data to produce the kriging-interpolated sections. Data coverage and standard deviations for the horizontal climatological sections are shown in Fig. A2 and Fig. A1 and for the vertical climatological sections in Fig. A4 and Fig. A3.
Vertical climatological sections of summertime (July–October) temperature and salinity over the periods 1930–2000 and 2000–2019 were made from hydrographic profiles within $\pm 0.15°$ perpendicular distance from an along-section from Storfjordrenna across the saddle and into the Olga Basin, comprised by the bin centres shown in Fig. 3a to where the profiles were first bin-averaged. The bin sizes in the along-section direction were the half between two neighboring bin centres. At the end points of the sections, profiles within $0.3°$ west (east) of the western (eastern) bin centre were included in the bin average. Each
profile was weighted with its distance to the bin centre to produce weighted bin-averaged sections with $5\,\mathrm{m}$ vertical resolution. The weighted bin-averaged sections were then interpolated onto a $500\,\mathrm{m}$ horizontal times $5\,\mathrm{m}$ vertical grid resolution using kriging interpolation.

Similarly, weighted bin-averaged horizontal distributions of depth-averaged temperature and salinity over the whole water column were estimated on a $0.5°$ longitude times $0.2°$ latitude grid resolution for the area $20°\,\mathrm{E}$–$28°\,\mathrm{E}$; $76°\,\mathrm{N}$–$78°\,\mathrm{N}$, where
each profile was weighted with its distance from the bin centre it belonged to. The weighted bin-averaged depth-averaged temperature and salinity horizontal sections were then interpolated onto a 2 times finer grid using the kriging interpolation method described above. In both grids the land points were excluded.



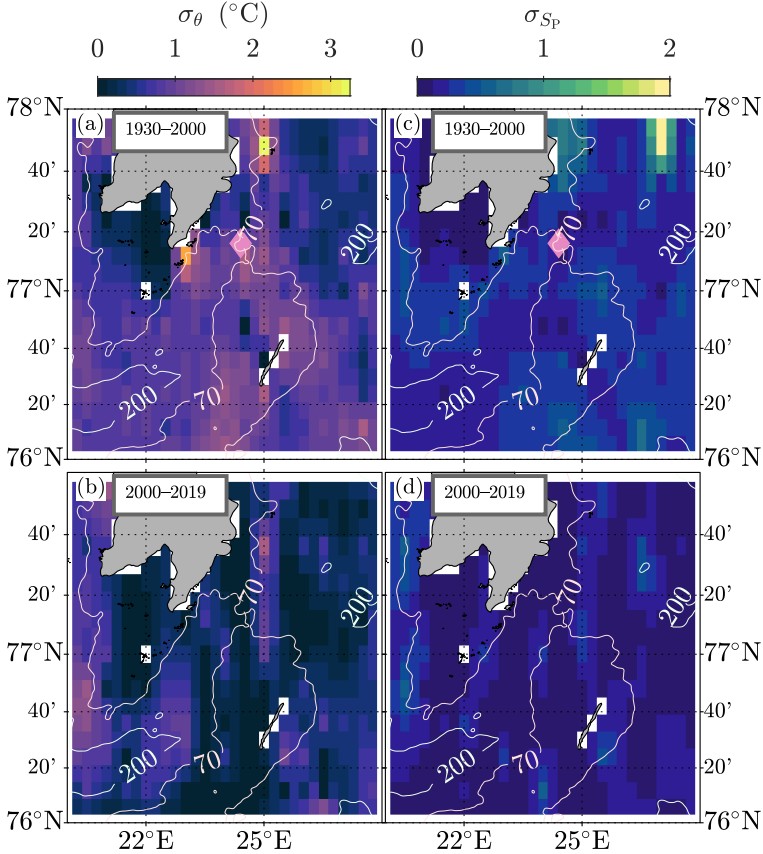

**Figure A1.** Standard deviations of the climatological horizontal hydrographic depth-averaged maps (Fig. 3). a–b) Standard deviation of potential temperature $\sigma_\theta$ (°C), c–d) standard deviation of practical salinity $\sigma_{S_P}$. a, c) 1930–2000, b, d) 2000–2019. The 70 m and 200 m isobaths are shown in white contours.

*Author contributions.* RS designed the study. KK and IF processed the mooring current velocity data. KK processed the mooring hydrographic data and the recent autumn and winter hydrographic data. RS compiled historical hydrographic data and made the climatological sections. KK analysed the data, prepared the figures and drafted the manuscript. RS wrote the text in the appendix. All authors have contributed to discussing the material and giving input and feedback on the analysis and the manuscript in several stages.

*Competing interests.* At least one of the (co-)authors is a member of the editorial board of Ocean Science.

*Acknowledgements.* This work was funded by the Research Council of Norway through the project The Nansen Legacy (RCN # 276730). The authors are grateful for the cooperation with the crew and the scientific/technical colleagues onboard the research vessels *Kronprins Haakon* and *G. O. Sars*. We thank colleagues who have read the manuscript, discussed the material, and provided feedback on the study.



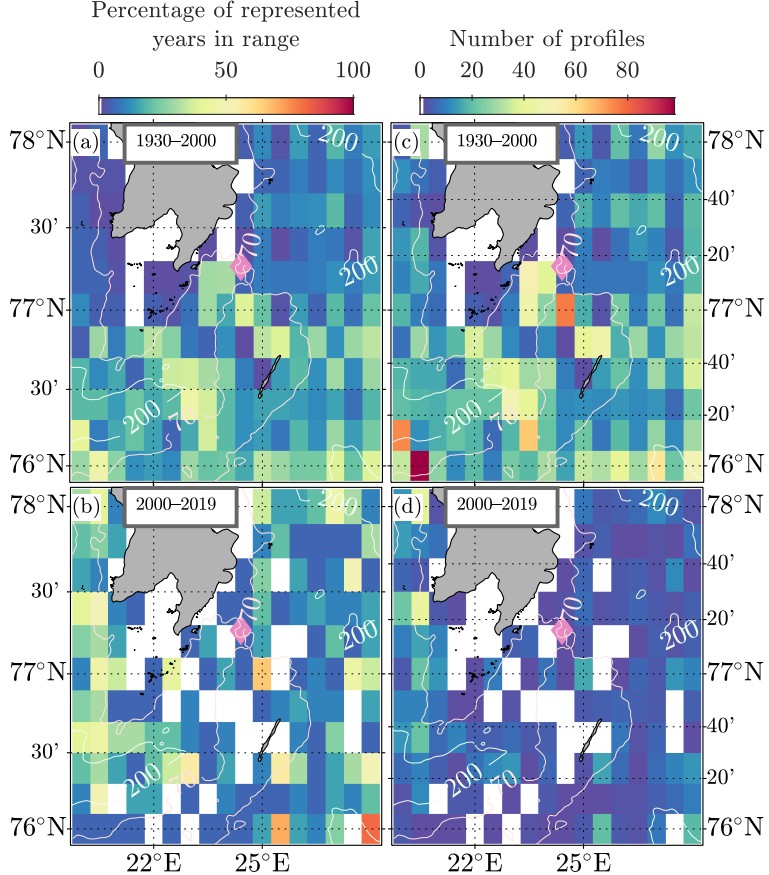

**Figure A2.** Data coverage of the climatological horizontal hydrographic depth-averaged maps (Fig. 3). Percentage of years with at least one profile in a) 1930–2000, b) 2000–2019, and number of profiles in c) 1930–2000, d) 2000–2019. The 70 m and 200 m isobaths are shown in white contours.

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



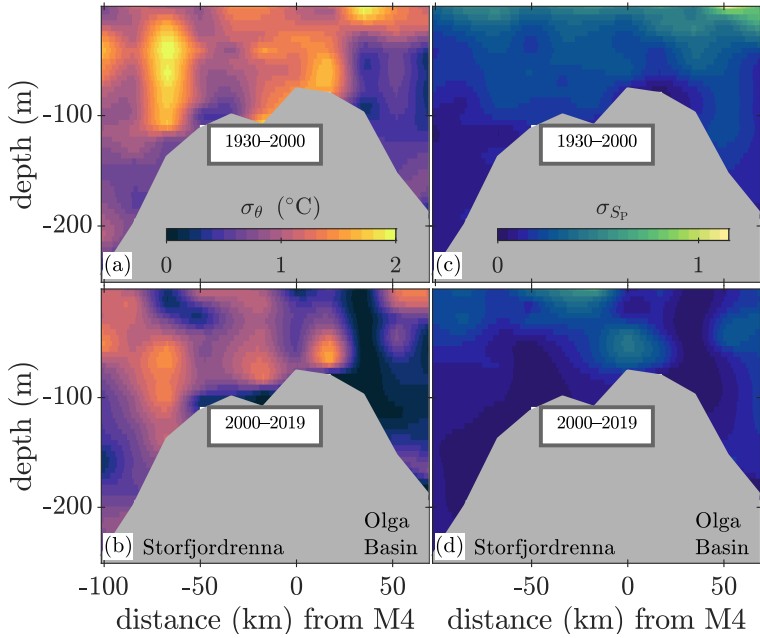

**Figure A3.** Standard deviations of the climatological vertical hydrographic transects (Fig. 4). a–b) Standard deviation of potential temperature $\sigma_\theta$ (°C), c–d) standard deviation of practical salinity $\sigma_{S_P}$. a, c) 1930–2000, b, d) 2000–2019. Horizontal axis shows distance to the mooring position, with positive values towards the Olga Basin. The location of the section is indicated with green dots in Fig. 3a.

Asbjørnsen, H., Årthun, M., Skagseth, Ø., and Eldevik, T.: Mechanisms Underlying Recent Arctic Atlantification, Geophysical Research Letters, 47, e2020GL088 036, https://doi.org/10.1029/2020GL088036, 2020.

Barton, B. I., Lenn, Y.-D., and Lique, C.: Observed Atlantification of the Barents Sea Causes the Polar Front to Limit the Expansion of Winter Sea Ice, Journal of Physical Oceanography, 48, 1849–1866, https://doi.org/10.1175/jpo-d-18-0003.1, 2018.

Bayoumy, M., Nilsen, F., and Skogseth, R.: Marine Heatwaves Characteristics in the Barents Sea Based on High Resolution Satellite Data (1982–2020), Frontiers in Marine Science, 9, 2022a.

Bayoumy, M., Nilsen, F., and Skogseth, R.: Interannual and Decadal Variability of Sea Surface Temperature and Sea Ice Concentration in the Barents Sea, Remote Sensing, 14, 4413, https://doi.org/10.3390/rs14174413, 2022b.

Bloshkina, E. V., Pavlov, A. K., and Filchuk, K.: Warming of Atlantic Water in Three West Spitsbergen Fjords: Recent Patterns and Century-

Long Trends, Polar Research, 40, https://doi.org/10.33265/polar.v40.5392, 2021.

Brown, N. J., Mauritzen, C., Li, C., Madonna, E., Isachsen, P. E., and LaCasce, J. H.: Rapid Response of the Norwegian Atlantic Slope Current to Wind Forcing, Journal of Physical Oceanography, 53, 389–408, https://doi.org/10.1175/JPO-D-22-0014.1, 2023.

Dalpadado, P., Arrigo, K. R., van Dijken, G. L., Skjoldal, H. R., Bagøien, E., Dolgov, A. V., Prokopchuk, I. P., and Sperfeld, E.: Climate Effects on Temporal and Spatial Dynamics of Phytoplankton and Zooplankton in the Barents Sea, Progress in Oceanography, 185, 102 320,

https://doi.org/10.1016/j.pocean.2020.102320, 2020.

Dickson, RR., Midttun, L., and Mukhin, AI.: The Hydrographic Conditions in the Barents Sea in August–September 1965–1968, International 0-group fish surveys in the Barents Sea, 18, 3–24, https://doi.org/10.17895/ices.pub.8051, 1970.



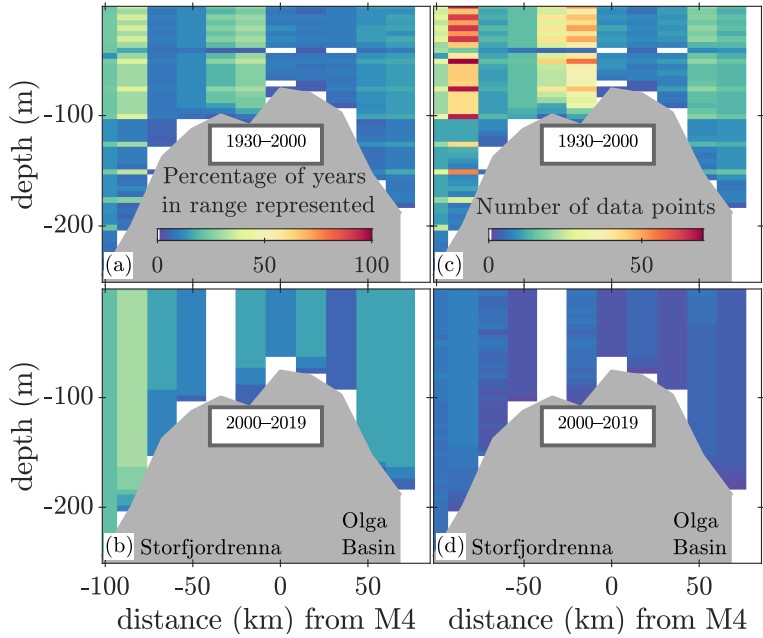

**Figure A4.** Data coverage of the climatological vertical hydrographic transects (Fig. 4). a–b) Percentage of years represented in a) 1930–2000, b) 2000–2019, and number of data points in c) 1930–2000, d) 2000–2019. Horizontal axis shows distance to the mooring position, with positive values towards the Olga Basin. The location of the section is indicated with green dots in Fig. 3a.

Eriksen, E., Skjoldal, H. R., Gjøsæter, H., and Primicerio, R.: Spatial and Temporal Changes in the Barents Sea Pelagic Compartment during the Recent Warming, Progress in Oceanography, 151, 206–226, https://doi.org/10.1016/j.pocean.2016.12.009, 2017.

Eriksen, E., Gjøsæter, H., Prozorkevich, D., Shamray, E., Dolgov, A., Skern-Mauritzen, M., Stiansen, J. E., Kovalev, Y., and Sunnanå, K.: From Single Species Surveys towards Monitoring of the Barents Sea Ecosystem, Progress in Oceanography, 166, 4–14, https://doi.org/10.1016/j.pocean.2017.09.007, 2018.

Erofeeva, S. and Egbert, G.: Arc5km2018: Arctic Ocean Inverse Tide Model on a 5 Kilometer Grid, 2018, https://doi.org/10.18739/A21R6N14K, 2020.

Fer, I., Skogseth, R., Astad, S. S., Baumann, T., Elliott, F., Falck, E., Gawinski, C., and Kolås, E. H.: SS-MSC2 Process Cruise/Mooring Service 2020: Cruise Report, The Nansen Legacy Report Series, https://doi.org/10.7557/nlrs.5798, 2021.

Geoffroy, M., Berge, J., Majaneva, S., Johnsen, G., Langbehn, T. J., Cottier, F., Mogstad, A. A., Zolich, A., and Last, K.: Increased Occurrence of the Jellyfish Periphylla Periphylla in the European High Arctic, Polar Biology, 41, 2615–2619, https://doi.org/10.1007/s00300-018-2368-4, 2018.

Gerland, S., Ingvaldsen, R. B., Reigstad, M., Sundfjord, A., Bogstad, B., Chierici, M., Hop, H., Renaud, P. E., Smedsrud, L. H., Stige, L. C., Årthun, M., Berge, J., Bluhm, B. A., Borgå, K., Bratbak, G., Divine, D. V., Eldevik, T., Eriksen, E., Fer, I., Fransson, A., Gradinger, R., Granskog, M. A., Haug, T., Husum, K., Johnsen, G., Jonassen, M. O., Jørgensen, L. L., Kristiansen, S., Larsen, A., Lien, V. S., Lind, S., Lindstrøm, U., Mauritzen, C., Melsom, A., Mernild, S. H., Müller, M., Nilsen, F., Primicerio, R., Søreide, J. E.,



van der Meeren, G. I., and Wassmann, P.: Still Arctic?—The Changing Barents Sea, Elementa: Science of the Anthropocene, 11, 00 088,
https://doi.org/10.1525/elementa.2022.00088, 2023.

Gjevik, B., Nøst, E., and Straume, T.: Model Simulations of the Tides in the Barents Sea, Journal of Geophysical Research, 99, 3337,
         https://doi.org/10.1029/93JC02743, 1994.

Good, S., Fiedler, E., Mao, C., Martin, M. J., Maycock, A., Reid, R., Roberts-Jones, J., Searle, T., Waters, J., While, J., and Worsfold, M.: The
         Current Configuration of the OSTIA System for Operational Production of Foundation Sea Surface Temperature and Ice Concentration
Analyses, Remote Sensing, 12, 720, https://doi.org/10.3390/rs12040720, 2020.

Guo, C., Ilicak, M., Fer, I., Darelius, E., and Bentsen, M.: Baroclinic Instability of the Faroe Bank Channel Overflow, Journal of Physical
         Oceanography, 44, 2698–2717, https://doi.org/10.1175/JPO-D-14-0080.1, 2014.

Häkkinen, S. and Cavalieri, D. J.: A Study of Oceanic Surface Heat Fluxes in the Greenland, Norwegian, and Barents Seas, Journal of
         Geophysical Research: Oceans, 94, 6145–6157, https://doi.org/10.1029/JC094iC05p06145, 1989.

Harms, I. H.: A Numerical Study of the Barotropic Circulation in the Barents and Kara Seas, Continental Shelf Research, 12, 1043–1058,
         https://doi.org/10.1016/0278-4343(92)90015-C, 1992.

Hersbach, H., Bell, B., Berrisford, P., Hirahara, S., Horányi, A., Muñoz-Sabater, J., Nicolas, J., Peubey, C., Radu, R., Schepers, D., Sim-
         mons, A., Soci, C., Abdalla, S., Abellan, X., Balsamo, G., Bechtold, P., Biavati, G., Bidlot, J., Bonavita, M., De Chiara, G., Dahlgren,
         P., Dee, D., Diamantakis, M., Dragani, R., Flemming, J., Forbes, R., Fuentes, M., Geer, A., Haimberger, L., Healy, S., Hogan, R. J.,
Hólm, E., Janisková, M., Keeley, S., Laloyaux, P., Lopez, P., Lupu, C., Radnoti, G., de Rosnay, P., Rozum, I., Vamborg, F., Vil-
         laume, S., and Thépaut, J.-N.: The ERA5 Global Reanalysis, Quarterly Journal of the Royal Meteorological Society, 146, 1999–2049,
         https://doi.org/10.1002/qj.3803, 2020.

Ingvaldsen, R. B., Assmann, K. M., Primicerio, R., Fossheim, M., Polyakov, I. V., and Dolgov, A. V.: Physical Manifestations and Ecological
         Implications of Arctic Atlantification, Nature Reviews Earth & Environment, 2, 874–889, https://doi.org/10.1038/s43017-021-00228-x,
585      2021.

Isaksen, K., Nordli, Ø., Ivanov, B., Køltzow, M. A. Ø., Aaboe, S., Gjelten, H. M., Mezghani, A., Eastwood, S., Førland, E., Benestad, R. E.,
         Hanssen-Bauer, I., Brækkan, R., Sviashchennikov, P., Demin, V., Revina, A., and Karandasheva, T.: Exceptional Warming over the Barents
         Area, Scientific Reports, 12, 9371, https://doi.org/10.1038/s41598-022-13568-5, 2022.

Jakobsson, M., Mayer, L., Coakley, B., Dowdeswell, J. A., Forbes, S., Fridman, B., Hodnesdal, H., Noormets, R., Pedersen, R., Rebesco, M.,
Schenke, H. W., Zarayskaya, Y., Accettella, D., Armstrong, A., Anderson, R. M., Bienhoff, P., Camerlenghi, A., Church, I., Edwards, M.,
         Gardner, J. V., Hall, J. K., Hell, B., Hestvik, O., Kristoffersen, Y., Marcussen, C., Mohammad, R., Mosher, D., Nghiem, S. V., Pedrosa,
         M. T., Travaglini, P. G., and Weatherall, P.: The International Bathymetric Chart of the Arctic Ocean (IBCAO) Version 3.0, Geophysical
         Research Letters, 39, n/a–n/a, https://doi.org/10.1029/2012GL052219, 2012.

Jakobsson, M., Mayer, L. A., Bringensparr, C., Castro, C. F., Mohammad, R., Johnson, P., Ketter, T., Accettella, D., Amblas, D., An, L., Arndt,
J. E., Canals, M., Casamor, J. L., Chauché, N., Coakley, B., Danielson, S., Demarte, M., Dickson, M. L., Dorschel, B., Dowdeswell, J. A.,
         Dreutter, S., Fremand, A. C., Gallant, D., Hall, J. K., Hehemann, L., Hodnesdal, H., Hong, J., Ivaldi, R., Kane, E., Klaucke, I., Krawczyk,
         D. W., Kristoffersen, Y., Kuipers, B. R., Millan, R., Masetti, G., Morlighem, M., Noormets, R., Prescott, M. M., Rebesco, M., Rignot, E.,
         Semiletov, I., Tate, A. J., Travaglini, P., Velicogna, I., Weatherall, P., Weinrebe, W., Willis, J. K., Wood, M., Zarayskaya, Y., Zhang, T.,
         Zimmermann, M., and Zinglersen, K. B.: The International Bathymetric Chart of the Arctic Ocean Version 4.0, Scientific Data, 7, 1–14,
https://doi.org/10.1038/s41597-020-0520-9, 2020.





Knipowitsch, N.: Hydrologische Untersuchungen Im Europäischen Eismeer, Annalen der Hydrographie und Maritimen Meteorologie, 33, 241–260, 1905.

Kohlbach, D., Goraguer, L., Bodur, Y., Müller, O., Amargant Arumí, M., Blix, K., Bratbak, G., Chierici, M., Dąbrowska, A., Dietrich, U., Edvardsen, B., García, L., Gradinger, R., Hop, H., Jones, E., Lundesgaard, Ø., Olsen, L., Reigstad, M., Saubrekka, K., and Assmy, P.:

Earlier Sea-Ice Melt Extends the Oligotrophic Summer Period in the Barents Sea with Low Algal Biomass and Associated Low Vertical Flux, Progress in Oceanography, p. 103018, https://doi.org/10.1016/j.pocean.2023.103018, 2023.

Kolås, E. H., Baumann, T. M., Skogseth, R., Koenig, Z., and Fer, I.: Western Barents Sea Circulation and Hydrography, Past and Present, https://doi.org/10.22541/essoar.169203078.81082540/v1, 2023.

Kowalik, Z. and Marchenko, A.: Tidal Motion Enhancement on Spitsbergen Bank, Barents Sea, Journal of Geophysical Research: Oceans,

128, e2022JC018 539, https://doi.org/10.1029/2022JC018539, 2023.

Kowalik, Z. and Proshutinsky, A. Yu.: Topographic Enhancement of Tidal Motion in the Western Barents Sea, Journal of Geophysical Research: Oceans, 100, 2613–2637, https://doi.org/10.1029/94JC02838, 1995.

Lewis, K. M., van Dijken, G. L., and Arrigo, K. R.: Changes in Phytoplankton Concentration Now Drive Increased Arctic Ocean Primary Production, Science, 369, 198–202, https://doi.org/10.1126/science.aay8380, 2020.

Lilly, J. M. and Olhede, S. C.: Wavelet Ridge Estimation of Jointly Modulated Multivariate Oscillations, in: 2009 Conference Record of the Forty-Third Asilomar Conference on Signals, Systems and Computers, pp. 452–456, IEEE, Pacific Grove, CA, USA, ISBN 978-1-4244-5825-7, https://doi.org/10.1109/ACSSC.2009.5469858, 2009.

Lind, S. and Ingvaldsen, R. B.: Variability and Impacts of Atlantic Water Entering the Barents Sea from the North, Deep Sea Research Part I: Oceanographic Research Papers, 62, 70–88, https://doi.org/10.1016/j.dsr.2011.12.007, 2012.

Lind, S., Ingvaldsen, R. B., and Furevik, T.: Arctic Warming Hotspot in the Northern Barents Sea Linked to Declining Sea-Ice Import, Nature Climate Change, 8, 634–639, https://doi.org/10.1038/s41558-018-0205-y, 2018.

Loeng, H.: Features of the Physical Oceanographic Conditions of the Barents Sea, Polar Research, 10, 5–18, https://doi.org/10.1111/j.1751-8369.1991.tb00630.x, 1991.

Lundesgaard, Ø., Sundfjord, A., Lind, S., Nilsen, F., and Renner, A. H. H.: Import of Atlantic Water and Sea Ice Controls the Ocean

Environment in the Northern Barents Sea, Ocean Science, 18, 1389–1418, https://doi.org/10.5194/os-18-1389-2022, 2022.

Lüpkes, C. and Birnbaum, G.: 'Surface Drag in the Arctic Marginal Sea-ice Zone: A Comparison of Different Parameterisation Concepts', Boundary-Layer Meteorology, 117, 179–211, https://doi.org/10.1007/s10546-005-1445-8, 2005.

Marchenko, A. and Kowalik, Z.: Tidal Wave–Elliptic Island Interaction above the Critical Latitude, Journal of Physical Oceanography, 53, 683–698, https://doi.org/10.1175/JPO-D-22-0018.1, 2023.

Marshall, J. and Shutts, G.: A Note on Rotational and Divergent Eddy Fluxes, Journal of Physical Oceanography, 11, 1677–1680, https://doi.org/10.1175/1520-0485(1981)011<1677:ANORAD>2.0.CO;2, 1981.

McClimans, T. A. and Nilsen, J. H.: Laboratory Simulation of the Ocean Currents in the Barents Sea, Dynamics of Atmospheres and Oceans, 19, 3–25, https://doi.org/10.1016/0377-0265(93)90030-B, 1993.

Mcdougall, T. J. and Krzysik, O. A.: Spiciness, Journal of Marine Research, 73, 141–152, 2015.

Midttun, L.: Formation of Dense Bottom Water in the Barents Sea, Deep Sea Research Part A. Oceanographic Research Papers, 32, 1233–1241, https://doi.org/10.1016/0198-0149(85)90006-8, 1985.

Nilsen, F., Skogseth, R., Vaardal-Lunde, J., and Inall, M.: A Simple Shelf Circulation Model: Intrusion of Atlantic Water on the West Spitsbergen Shelf, Journal of Physical Oceanography, 46, 1209–1230, https://doi.org/10.1175/JPO-D-15-0058.1, 2016.





Olhede, S. and Walden, A.: Generalized Morse Wavelets, IEEE Transactions on Signal Processing, 50, 2661–2670,
https://doi.org/10.1109/TSP.2002.804066, 2002.

Onarheim, I. H. and Årthun, M.: Toward an Ice-Free Barents Sea, Geophysical Research Letters, 44, 8387–8395,
https://doi.org/10.1002/2017GL074304, 2017.

OSTIA: Global Ocean OSTIA Sea Surface Temperature and Sea Ice Reprocessed, https://doi.org/10.48670/moi-00168, 2023.

Oziel, L., Sirven, J., and Gascard, J.-C.: The Barents Sea Frontal Zones and Water Masses Variability (1980–2011), Ocean Science, 12,
169–184, https://doi.org/10.5194/os-12-169-2016, 2016.

Oziel, L., Baudena, A., Ardyna, M., Massicotte, P., Randelhoff, A., Sallée, J.-B., Ingvaldsen, R. B., Devred, E., and Babin, M.: Faster
Atlantic Currents Drive Poleward Expansion of Temperate Phytoplankton in the Arctic Ocean, Nature Communications, 11, 1705,
https://doi.org/10.1038/s41467-020-15485-5, 2020.

Pavlov, A. K., Tverberg, V., Ivanov, B. V., Nilsen, F., Falk-Petersen, S., and Granskog, M. A.: Warming of Atlantic Water in Two West
Spitsbergen Fjords over the Last Century (1912-2009), Polar Research, 32, 1–14, https://doi.org/10.3402/polar.v32i0.11206, 2013.

Percival, D. B. and Walden, A. T.: Spectral Analysis for Physical Applications, Cambridge University Press, ISBN 978-0-521-35532-2,
https://doi.org/10.1017/CBO9780511622762, 1993.

Polyakov, I. V., Pnyushkov, A. V., Alkire, M. B., Ashik, I. M., Baumann, T. M., Carmack, E. C., Goszczko, I., Guthrie, J., Ivanov, V. V.,
Kanzow, T., Krishfield, R., Kwok, R., Sundfjord, A., Morison, J., Rember, R., and Yulin, A.: Greater Role for Atlantic Inflows on Sea-Ice
Loss in the Eurasian Basin of the Arctic Ocean, Science, 356, 285–291, https://doi.org/10.1126/science.aai8204, 2017.

Quadfasel, D., Rudelst, B., and Kurz, K.: Outflow of Dense Water from a Svalbard Fjord into the Fram Strait, Deep-Sea Research, 35,
1143–1150, 1988.

Renner, A. and Sundfjord, A.: Mooring Service Cruise 2021: Cruise Report, The Nansen Legacy Report Series,
https://doi.org/10.7557/nlrs.6461, 2022.

Rieke, O., Årthun, M., and Dörr, J. S.: Rapid Sea Ice Changes in the Future Barents Sea, The Cryosphere, 17, 1445–1456,
https://doi.org/10.5194/tc-17-1445-2023, 2023.

Rudels, B., Jones, E. P., Anderson, L. G., and Kattner, G.: On the Intermediate Depth Waters of the Arctic Ocean, in: The Polar Oceans
and Their Role in Shaping the Global Environment, pp. 33–46, American Geophysical Union (AGU), ISBN 978-1-118-66388-2,
https://doi.org/10.1029/GM085p0033, 1994.

Schauer, U.: The Release of Brine-Enriched Shelf Water from Storfjord into the Norwegian Sea, Journal of Geophysical Research: Oceans,
100, 16 015–16 028, https://doi.org/10.1029/95JC01184, 1995.

Schauer, U., Muench, R. D., Rudels, B., and Timokhov, L.: Impact of Eastern Arctic Shelf Waters on the Nansen Basin Intermediate Layers,
Journal of Geophysical Research: Oceans, 102, 3371–3382, https://doi.org/10.1029/96JC03366, 1997.

Schauer, U., Loeng, H., Rudels, B., Ozhigin, V. K., and Dieck, W.: Atlantic Water Flow through the Barents and Kara Seas, Deep Sea
Research Part I: Oceanographic Research Papers, 49, 2281–2298, https://doi.org/10.1016/S0967-0637(02)00125-5, 2002.

Shu, Q., Wang, Q., Song, Z., and Qiao, F.: The Poleward Enhanced Arctic Ocean Cooling Machine in a Warming Climate, Nature Commu-
nications, 12, 2966, https://doi.org/10.1038/s41467-021-23321-7, 2021.

Skagseth, Ø., Eldevik, T., Årthun, M., Asbjørnsen, H., Lien, V. S., and Smedsrud, L. H.: Reduced Efficiency of the Barents Sea Cooling
Machine, Nature Climate Change, 10, 661–666, https://doi.org/10.1038/s41558-020-0772-6, 2020.

Skogseth, R., Haugan, P. M., and Haarpaintner, J.: Ice and Brine Production in Storfjorden from Four Winters of Satellite and in Situ
Observations and Modeling, Journal of Geophysical Research C: Oceans, 109, 1–15, https://doi.org/10.1029/2004JC002384, 2004.





Skogseth, R., McPhee, M. G., Nilsen, F., and Smedsrud, L. H.: Creation and Tidal Advection of a Cold Salinity Front in Storfjorden: 1. Polynya Dynamics, Journal of Geophysical Research: Oceans, 118, 3278–3291, https://doi.org/10.1002/jgrc.20231, 2013.

Skogseth, R., Ellingsen, P., Berge, J., Cottier, F. R., Falk-Petersen, S., Ivanov, B. V., Nilsen, F., Søreide, J. E., and Vader, A.: UNIS Hydrographic Database, https://doi.org/10.21334/unis-hydrography, 2019.

Skogseth, R., Olivier, L. L., Nilsen, F., Falck, E., Fraser, N. J., Tverberg, V., Ledang, A. B., Vader, A., Jonassen, M. O., Søreide, J., Cottier, F., Berge, J., Ivanov, B. V., and Falk-Petersen, S.: Variability and Decadal Trends in the Isfjorden (Svalbard) Ocean Climate and Circulation – An Indicator for Climate Change in the European Arctic, Progress in Oceanography, 187, 102 394, https://doi.org/10.1016/j.pocean.2020.102394, 2020.

Slepian, D.: Prolate Spheroidal Wave Functions, Fourier Analysis, and Uncertainty — V: The Discrete Case, The Bell System Technical Journal, 57, 1371–1430, https://doi.org/10.1002/j.1538-7305.1978.tb02104.x, 1978.

Smedsrud, L. H., Esau, I., Ingvaldsen, R. B., Eldevik, T., Haugan, P. M., Li, C., Lien, V. S., Olsen, A., Omar, A. M., Risebrobakken, B., Sandø, A. B., Semenov, V. A., and Sorokina, S. A.: The Role of the Barents Sea in the Arctic Climate System, Reviews of Geophysics, 51, 415–449, https://doi.org/10.1002/rog.20017, 2013.

Smedsrud, L. H., Muilwijk, M., Brakstad, A., Madonna, E., Lauvset, S. K., Spensberger, C., Born, A., Eldevik, T., Drange, H., Jeansson, E., Li, C., Olsen, A., Skagseth, Ø., Slater, D. A., Straneo, F., Våge, K., and Årthun, M.: Nordic Seas Heat Loss, Atlantic Inflow, and Arctic Sea Ice Cover Over the Last Century, Reviews of Geophysics, 60, e2020RG000 725, https://doi.org/10.1029/2020RG000725, 2022.

Strzelewicz, A., Przyborska, A., and Walczowski, W.: Increased Presence of Atlantic Water on the Shelf South-West of Spitsbergen with Implications for the Arctic Fjord Hornsund, Progress in Oceanography, 200, 102 714, https://doi.org/10.1016/j.pocean.2021.102714, 2022.

Sundfjord, A. and Renner, A.: Mooring Service Cruise 2019: Cruise Report, The Nansen Legacy Report Series, https://doi.org/10.7557/nlrs.5797, 2021.

Thomson, D.: Spectrum Estimation and Harmonic Analysis, Proceedings of the IEEE, 70, 1055–1096, https://doi.org/10.1109/PROC.1982.12433, 1982.

Tverberg, V., Skogseth, R., Cottier, F., Sundfjord, A., Walczowski, W., Inall, M. E., Falck, E., Pavlova, O., and Nilsen, F.: The Kongsfjorden Transect: Seasonal and Inter-annual Variability in Hydrography, in: The Ecosystem of Kongsfjorden, Svalbard, edited by Hop, H. and Wiencke, C., Advances in Polar Ecology, pp. 49–104, Springer International Publishing, Cham, ISBN 978-3-319-46425-1, https://doi.org/10.1007/978-3-319-46425-1_3, 2019.

Vihtakari, M., Sundfjord, A., and de Steur, L.: Barents Sea Ocean-Current Arrows Modified from Eriksen et al. (2018), 2019.

Vinje, T., Jensen, H., Johnsen, A. S., Løset, S., Hamran, S. E., Løvaas, S. M., and Erlingson, B.: IDAP-89 R/V Lance Deployment. Vol. 2. Field Observations and Analysis, Tech. rep., Norwegian Polar Institute/SINTEF NHL, Oslo/Trondheim, 1989.

Vivier, F., Lourenço, A., Michel, E., Skogseth, R., Rousset, C., Lansard, B., Bouruet-Aubertot, P., Boutin, J., Bombled, B., Cuypers, Y., Crispi, O., Dausse, D., Le Goff, H., Madec, G., Vancoppenolle, M., Van der Linden, F., and Waelbroeck, C.: Summer Hydrography and Circulation in Storfjorden, Svalbard, Following a Record Low Winter Sea-Ice Extent in the Barents Sea, Journal of Geophysical Research: Oceans, 128, e2022JC018 648, https://doi.org/10.1029/2022JC018648, 2023.

Wickström, S., Jonassen, M. O., Vihma, T., and Uotila, P.: Trends in Cyclones in the High-Latitude North Atlantic during 1979–2016, Quarterly Journal of the Royal Meteorological Society, 146, 762–779, https://doi.org/10.1002/qj.3707, 2020.

Wold, A., Hop, H., Svensen, C., Søreide, J. E., Assmann, K. M., Ormanczyk, M., and Kwasniewski, S.: Atlantification Influences Zooplankton Communities Seasonally in the Northern Barents Sea and Arctic Ocean, Progress in Oceanography, 219, 103 133, https://doi.org/10.1016/j.pocean.2023.103133, 2023.