# Peer review of "An emerging pathway of Atlantic Water to the Barents Sea through the Svalbard Archipelago: drivers and variability"

_EGUsphere, 2023_

## Author Comment (AC1)

**Response to Reviewer 1's comments**

We thank Anonymous Referee #1 for their review of our manuscript. In this response letter, the reviewer's comment is written in black Arial and our response is in purple (this font style). Our final response will include more details together with a new version of the manuscript with tracked changes not provided here.

**Summary**

In this manuscript the authors suggest an existence of newly emerging pathway for the warm and saline Atlantic Water entering the Arctic Ocean via the northwestern Barents Sea. They propose that the new corridor is located between Edgeøya and Hopen, the islands situated southeast of Spitsbergen, in the easternmost, shallow part of Storfjordrenna. The authors present and thoroughly analyse recently collected observations, including a year-long time series from moored instruments spanning 2018-2019, as well as three hydrographic transects from ships during the autumns of 2019-2021. The obtained results are put in a wider climatological context drawing upon a large database of quality-controlled observations spanning nearly a century, from 1930 to 2019. Although the available database primarily covers summer seasons, it provides sufficient information to reconstruct the climatological vertical hydrographic section near the mooring location.

**General assessment**

The manuscript's topic aligns well with the scope of 'Ocean Science 'and addresses the evolving role of the Barents Sea in the context of the warming Arctic and changing ocean climate, a subject of extensive discussion within the scientific community in recent years. Despite the somewhat limited dataset in terms of both time and spatial coverage (comprising a single shallow mooring and relatively short hydrographic surveys obtained in autumn), the authors made an effort to comprehend and characterise the variability occurring across various time scales in meticulous detail. Through the application of various filtering techniques, the authors have successfully distinguished between slow-acting and fast-acting mechanisms underlying the oscillating currents and observed changes in water temperature. In my opinion, the manuscript is written and illustrated well enough and makes a reasonable research statement in the previously undocumented area. However, I also found it slightly overloaded with content and details and thus difficult to follow at times. Therefore, I suggest

that it can be published after some minor corrections.

Thank you for the thorough and detailed review and the constructive feedback. We are very pleased to read that our manuscript was well-received. In the revised version, we reduced the details not central to the paper and streamlined the content for an easier reading. Please see below where we address your specific comments.

**Comments and remarks:**

Introduction:

Line 28-30: Please add some newer references since the mentioned, although crucial studies, are more than 20 years old. Incorporating some recent studies will help strengthen the manuscript's relevance and ensure that it remains up-to-date with the latest advancements in the field.

Thank you for pointing that out. We have added citations to several recent studies for this paragraph, including Rudels et al. (2013), Lind et al. (2018), Årthun et. al. (2011).

Line 59: 'Summer of 2016 had the warmest and longest-lasting marine heatwave in the Barents Sea' – you may add 'so far'.

We have added 'so far'.

Line 70: 'Here, we investigate the physical processes that drive and mediate the inflow of Atlantic-origin water masses' – I would say that the water masses are rather briefly classified in this paper, if at all. I wonder why: is it problematic to compare the CT and SA with previous classifications made for T&S? Or there is another reason behind it., i.e. you focus on mechanisms rather than structure of the water column itself? In this case, providing a brief overview of water masses may be sufficient (one-two sentences).

That is a good point – we did not use strict water mass classifications, as this is not central to our study. Using the relevant water mass classifications that are appropriate for the nearby regions would complicate the analysis and discussion without adding more insight. Classifying water masses using CT and SA instead of T and S is often done and is typically not an issue. We chose, in the end, to focus on the mechanisms and whether the water observed is of Atlantic origin, i.e. whether it is (modified or transformed) AW from Storfjordrenna that carries

heat into the colder domain of the Barents Sea. We have now added a definition of Atlantic Water (CT exceeding 2°C and SA exceeding 35.06 g/kg) and clarified what we mean by Atlantic-origin water by inserting "We define the Atlantic-origin water, as the AW that is modified or transformed en-route from the WSC through Storfjordrenna into relatively warm and saline water compared to the surrounding water masses but colder and less saline than pure AW."

**Data & methods**

Line 78: '70 meter deep saddle on Hopenbanken between Storfjordrenna and the interior of the Barents Sea' – describe here what 'saddle' is in bathymetry. Also, what 'banken' and 'renna' are (instead of indicating it only in the 'Results' section).

With "saddle" we refer to the crest of the sill in the bank between the trough Storfjordrenna and the interior of the Barents Sea (Olga Basin). We clarified it so that it reads 'on the shallow bank Hopenbanken between the trough Storfjordrenna and the interior of the Barents Sea (Olga Basin)'.

Line 95: 'the measured current were in fairly good agreement with Arc5km2018' – what does 'fairly good' mean here? Also, change 'current' for 'currents'.

We clarified by adding "i.e., ellipse inclinations agreed within 14 degrees and semimajor axes within 1 cm/s" and changed 'current' to 'currents'.

Line 99: 'Wind speed and direction in the region are taken from the ERA5 reanalyses' - have you considered or tried to use CARRA reanalysis? Any thoughts on this?

We tried both ERA5 and CARRA in preliminary analyses. Comparing them, we found that they agreed well. And considering our use case, we concluded that ERA5 had a sufficiently good temporal and spatial resolution for our analysis. We preferred ERA5 because it made the analysis quicker and easier as CARRA is a relatively heavy data set on a less convenient grid.

Line 120-121: 'the multitaper method from Percival and Walden (1993) with Slepian data tapers (e.g., Slepian, 1978; Thomson, 1982). To analyse the time-variability of the spectral components, wavelet transforms following Lilly and Olhede (2009) with generalised Morse wavelets (Olhede and Walden, 2002) were used' – adding a brief explanation about the general purpose of using the methods mentioned (rationale behind and how they contribute to the study's objectives) in the manuscript could enhance the reader's understanding and

contribute to smoother reading.

We agree. Our general purpose in using these methods was to identify different time scales associated with different processes. To clarify this, we changed the first sentence to 'In order to identify the time scales of variability, we used spectral analysis', and reformulated the last sentence to "The values for the parameters that define the Morse wavelets were chosen to be $\gamma = 3$ for symmetric wavelets and $\beta = 5$ to resolve the variability on time scales between semidiurnal and a few weeks reasonably in both frequency and time."

Line 134: 'bottom half of the water column' – why? Do you aim to separate the surface layer circulation or rather ADCP data quality away from transducer was a limiting factor here?

The ADCP data quality was quite good through most of the water column throughout the time series (the range shown in Figure 6d is the range with quality-controlled data). The choice of using the bottom half of the water column was due to having temperature only at the bottom. While the near-bottom temperature may not be representative for the whole water column when the water column is at times stratified in temperature, available CTD profiles suggest that the lower half of the water column is typically well mixed. We added a sentence ("The velocity data are depth-averaged over the bottom half of the water column to use with the temperature measurements that are available only near the bottom.") to this paragraph to explain this.

**Results:**

I advise the authors to consider prioritising the most significant results and simplifying the presentation of detailed analyses, particularly in Section 3.4. The abundance of details makes it challenging for readers to follow the discussion effectively. Instead of presenting all available analyses, the authors might consider using a simple table with numerical data to present these results. Additionally, referring large part of the results solely to Figure 8 can be difficult for readers.

Thank you for the constructive feedback. We agree that there are details in the text and figures (especially in Section 3.4 and Figure 8), which are not critical for our results and conclusions. We do agree that especially Section 3.4 and Figure 8 can be simplified. In the revised version we removed Figure 8, as the important parts are visible in other figures (Figures 7, 9, and 10), and shortened the text in Section 3.4.

Line 152: 'salinity has increased by 0.05 g/kg to 0.35g/kg'- it's interesting how 34.4 isohaline deepened, and that negative T anomaly is noticeable in this location as well (Fig 4b-c) in the last decades in the Olga Basin. Is it an effect of more intense mixing? Or perhaps this is a signal of the Polar Front sharpening and concurrent steepening of isohalines/isopycnals? As we know from 'Introduction' section (Lines 38-42) the amount of inflowing freshwater has decreased in the northern Barents Sea…

Thank you for pointing this out, which we didn't catch earlier. We now mention the deepening of the 34.4 isohaline toward the Olga Basin and the colocated negative T anomaly. In order to answer your question, a targeted analysis is needed. Having sparse data in OB, we are hesitant to interpret "details" or areas with especially sparse coverage and low standard deviation (due to coverage). In Section 4.4, we have a short discussion on the role of changing stratification in the Olga Basin.

Line 161: 'At this time, AW was present in Storfjordrenna and close to the mooring' – again, I miss the water mass classification in this manuscript. As the AW definition was not mentioned in the Introduction and Data & methods part, it's necessary to introduce it here before you start to describe the results. As you use CT and SA, it will be good to define AW in these scales.

We agree. In the Data and Methods section, we added the CT&SA thresholds we used for AW (CT $\geq 2°$C and SA $\geq$ 35.06 g/kg) and what we mean with "Atlantic-origin water" ("We define the Atlantic-origin water as the AW that is modified or transformed en-route from the WSC through Storfjordrenna into relatively warm and saline water compared to the surrounding water masses but colder and less saline than pure AW."). We also specified the AW properties observed in Storfjordrenna on this transect in the sentence "At this time, AW was present in Storfjordrenna and close to the mooring (not shown): 70 km southwest of the mooring, water with temperature 2.9 $°$C to 3.7 $°$C and salinity 34.84 g/kg to 35.02 g/kg was found at depth, which is within the temperature range for pure AW and slightly lower than the salinity threshold of 35.06 g/kg \citep{sundfjord2020}. Also relatively warm water of around 0.5 $°$C was situated at depth 10 km to the west of the mooring position."

Line 198: 'to 34 cm s−1 near the surface' - when exactly? It will be good to find it in the plot - one cannot see the local maxima near the surface.

In the beginning of March – we have clarified this and improved the readability of the paragraph.

Discussion:

Winds, ice, tides, and AW inflows as well as upstream conditions definitely work together to shape the seasonal and interannual variability. However, attempting to address all the complex factors in one comprehensive article is probably ambitious task. I found the discussion too broad, it's understandable given the extensive range of results presented in previous sections. However, I miss a clear statement regarding which mechanisms are deemed most responsible for the observed high interannual variability.

We have added in Conclusions that the semidiurnal tides lead half as much heat across-saddle compared to weather-band processes, and also added a sentence about the large-scale forcing and upstream conditions that likely are major drivers of the interannual variability.

Line 323: 'water column has not yet cooled down at that time' – indeed, the observed differences between mid-October and November suggest a potential regime shift from Atlantic Water (AW) dominance to Arctic Water (AW) dominance during that time of the year. But of course, it could also be an exceptional year with more/warmer inflowing AW as you wrote.

Fig. 5 indeed indicates a shift from Atlantic to Arctic Water dominance during that time of year due to atmospheric cooling and influence from the colder, fresher environment in northeast. However, our Fig. 12 suggests that for autumn 2020, there was a stronger inflow of Atlantic Water lasting into winter, unlike other years where November was either Arctic dominated or cooled down. Autumn 2018 stood out as warmer and saltier through October and November, with sea ice starting later (Figure 6). Lundesgaard et al. (2022) also observed this south of Kvitøya. We interpret this as, in addition to the normal seasonal cycle, interannual variability in Atlantic Water inflow to the northern Barents Sea substantially influences the study area.

We modified the discussion in the manuscript to better reflect this point and include a reference to Lundesgaard et al.

Figures:

In general, I think a few plots and subplots would be better with some additional information that would make them more self-explanatory. Increasing the thickness of lines, enlarging markers, and reducing clutter can improve the visibility of key data points. It feels like this manuscript has many more figures than only 12 (plus these in appendices).

Thank you for the constructive feedback. We agree with your suggestions and have adjusted the figures accordingly. We have also removed Figure 8 (see below).

Figure 3: isotherm 0 deg C (light grey) - too like 200 m isobath - can it be changed?

We agree. The isotherm is now blue and should be more distinguishable from the isobaths.

Figure 7f-g: Perhaps some simple legend naming the lines can be added to read these plots without checking back and forward with the caption?

We have added a legend and also given each curve a distinct colour. We now show both rotary components for both seasons in a shared panel to reduce the number of curves.

Figure 8: This plot seems a bit too complicated to me. Also, it would profit from adding some legends.

We have removed Figure 8 since it contained information that was not central to the manuscript. We agree with your suggestion above to prioritise the most significant results. The main results which we wanted to communicate through Figure 8 are also visible in Figures 7, 9, and 10. Furthermore, removing Figure 8 allows removing some text that describes this figure, and further simplifies the content for the reader.

Figure 9: Why is there no current anomalies arrow in the subplot c)? The reference unit vectors for wind and currents are missing, as well.

We have added reference vectors for wind stress (0.2 W/m^2) in panel b and for current (5 cm/a) in panel d.

In Fig. 9c, the composite average of the current anomaly is minuscule (about a hundredth of the current anomalies in panels b and d) because we have averaged over the non-extreme wind stress conditions. Panel c shows the spatial picture of the centre of the cross (standard deviation cross) for the pale blue dots in panel a, which represents weak wind stress from north/northeast (small negative Ekman transport) and a (very close to) zero current anomaly.

Figure 12: 'Sea ice concentration (black, left vertical axis) and sea surface temperature (red, right vertical axis)' - from which product - write it here, also - add years on the subplots - it would be easier to look at.

Thank you for this suggestion. We have added the years on the subplots and added that the

data are from OSTIA.

---

## Author Response (AR1)

**Response to reviewers**

We thank both Anonymous Referees for their thorough reviews and constructive feedback for our manuscript. Their suggestions have led to substantial improvements in the revised version of the manuscript.

Both reviewers found that the original manuscript was in some parts overloaded and sometimes difficult to follow. We implemented their concrete suggestions and found that they improved the clarity in both figures and text and helped reduce details that were less important for the key findings. Additionally, we have worked on improving the language and flow of the text throughout the manuscript, including using more active tense.

Please see below for all Referee comments together with our responses. The revised manuscript and a document tracking changes in the manuscript (latex-diff) are attached. The line numbers given in this response letter refer to the line numbers in the revised manuscript.

**Response to Reviewer 1's comments**

**Summary**

In this manuscript the authors suggest an existence of newly emerging pathway for the warm and saline Atlantic Water entering the Arctic Ocean via the northwestern Barents Sea. They propose that the new corridor is located between Edgeøya and Hopen, the islands situated southeast of Spitsbergen, in the easternmost, shallow part of Storfjordrenna. The authors present and thoroughly analyse recently collected observations, including a year-long time series from moored instruments spanning 2018-2019, as well as three hydrographic transects from ships during the autumns of 2019-2021. The obtained results are put in a wider climatological context drawing upon a large database of quality-controlled observations spanning nearly a century, from 1930 to 2019. Although the available database primarily covers summer seasons, it provides sufficient information to reconstruct the climatological vertical hydrographic section near the mooring location.

**General assessment**

The manuscript's topic aligns well with the scope of 'Ocean Science 'and addresses the evolving role of the Barents Sea in the context of the warming Arctic and changing ocean climate, a subject of extensive discussion within the scientific community in recent years. Despite the somewhat limited dataset in terms of both time and spatial coverage (comprising a single shallow mooring and relatively short hydrographic surveys obtained in autumn), the authors made an effort to comprehend and characterise the variability occurring across various time scales in meticulous detail. Through the application of various filtering techniques, the authors have successfully distinguished between slow-acting and fast-acting mechanisms underlying the oscillating currents

and observed changes in water temperature.

In my opinion, the manuscript is written and illustrated well enough and makes a reasonable research statement in the previously undocumented area. However, I also found it slightly overloaded with content and details and thus difficult to follow at times. Therefore, I suggest that it can be published after some minor corrections.

Thank you for the thorough and detailed review and the constructive feedback. We are very pleased to read that our manuscript was well-received. In the revised version, we reduced the details not central to the paper and streamlined the content for an easier reading. Please see below where we address your specific comments.

**Comments and remarks:**

*Introduction:*

Line 28-30: Please add some newer references since the mentioned, although crucial studies, are more than 20 years old. Incorporating some recent studies will help strengthen the manuscript's relevance and ensure that it remains up-to-date with the latest advancements in the field.

Thank you for pointing that out. We have included several more recent studies in this part of the paragraph (new references pasted below) so that it now reads:

"As AW flows through the Barents Sea, it is modified and transformed due to intense cooling through heat loss to the atmosphere (Häkkinen and Cavalieri, 1989; Årthun and Schrum, 2010; Lind et al., 2018; Ivanov et al., 2020), brine release from sea-ice growth (e.g., Schauer et al., 2002; Ivanov et al., 2020), and mixing with the cooler and fresher Arctic Water (Loeng, 1991; Lind et al., 2018) and denser brine-enriched waters (Schauer et al., 2002). The Barents Sea has been termed a "cooling machine" (Smedsrud et al., 2013) whereby AW is transformed into Barents Sea Water. It is an important area for dense water production (Knipowitsch, 1905; Midttun, 1985; Schauer et al., 2002; Årthun et al., 2011) and a source of intermediate waters to the Arctic Ocean (Rudels et al., 1994; Schauer et al., 1997, 2002; Rudels et al., 2013)." (Lines 24–30)

Årthun, M., Ingvaldsen, R. B., Smedsrud, L. H., and Schrum, C.: Dense Water Formation and Circulation in the Barents Sea, Deep-Sea Research Part I: Oceanographic Research Papers, 58, 801–817, https://doi.org/10.1016/j.dsr.2011.06.001, 2011.

Ivanov, V. V., Frolov, I. E., and Filchuk, K. V.: Transformation of Atlantic Water in the North-Eastern Barents Sea in Winter, Arctic and Antarctic Research, 66, 246–266, https://doi.org/10.30758/0555-2648-2020-66-3-246-266, 2020.

Lind, S., Ingvaldsen, R. B., and Furevik, T.: Arctic Warming Hotspot in the Northern Barents Sea Linked to Declining Sea-Ice Import, Nature Climate Change, 8, 634–639, https://doi.org/10.1038/s41558-018-0205-y, 2018.

Rudels, B., Schauer, U., Björk, G., Korhonen, M., Pisarev, S., Rabe, B., and Wisotzki, A.: Observations of Water Masses and Circulation with Focus on the Eurasian Basin of the Arctic Ocean from the 1990s to the Late 2000s, Ocean Science, 9, 147–169, https://doi.org/10.5194/os-9-147-2013, 2013.

Line 59: 'Summer of 2016 had the warmest and longest-lasting marine heatwave in the Barents Sea' – you may add 'so far'.

We have added 'so far'.

Line 70: 'Here, we investigate the physical processes that drive and mediate the inflow of Atlantic-origin water masses' – I would say that the water masses are rather briefly classified in this paper, if at all. I wonder why: is it problematic to compare the CT and SA with previous classifications made for T&S? Or there is another reason behind it., i.e. you focus on mechanisms rather than structure of the water column itself? In this case, providing a brief overview of water masses may be sufficient (one-two sentences).

That is a good point – we did not use strict water mass classifications, as this is not central to our study. Using the relevant water mass classifications that are appropriate for the nearby regions would complicate the analysis and discussion without adding more insight. Classifying water masses using CT and SA instead of T and S is often done and is typically not an issue. We chose, in the end, to focus on the mechanisms and whether the water observed is of Atlantic origin, i.e. whether it is (modified or transformed) AW from Storfjordrenna that carries heat into the colder domain of the Barents Sea. We have now added a definition of Atlantic Water and clarified what we mean by Atlantic-origin water in the Data and Methods section (Lines 116–119):

"AW is defined as water with Conservative Temperature $\Theta \geq 2 \circ C$ and Absolute Salinity SA $\geq 35.06$ g kg−1 (Sundfjord et al., 2020). We refer to the Atlantic-origin water as the AW that is modified or transformed en route from the WSC through Storfjordrenna into relatively warm and saline water compared to the surrounding water masses, but colder and less saline than pure AW."

*Data & methods*

Line 78: '70 meter deep saddle on Hopenbanken between Storfjordrenna and the interior of the Barents Sea' – describe here what 'saddle' is in bathymetry. Also, what 'banken' and 'renna' are (instead of indicating it only in the 'Results' section).

With "saddle" we refer to the crest of the sill in the bank between the trough Storfjordrenna and the interior the Barents Sea (Olga Basin). We clarified it so that it reads 'on the shallow bank Hopenbanken between the trough Storfjordrenna and the interior of the Barents Sea'.

We have also added some descriptive nouns in the Introduction section, e.g., "into the trough Storfjordrenna in the Svalbard Archipelago (Fig. 2), north of the BSO and the shallow bank Spitsbergenbanken" (line 53) and "the trench Hopendjupet" (line 66).

Line 95: 'the measured current were in fairly good agreement with Arc5km2018' – what does 'fairly good' mean here? Also, change 'current' for 'currents'.

We clarified by adding "i.e., ellipse inclinations agreed within 14 degrees and semimajor axes within 1 cm/s" and changed 'current' to 'currents'.

Line 99: 'Wind speed and direction in the region are taken from the ERA5 reanalyses' - have you considered or tried to use CARRA reanalysis? Any thoughts on this?

We tried both ERA5 and CARRA in preliminary analyses. Comparing them, we found that they agreed well. And considering our use case, we concluded that ERA5 had a sufficient temporal and spatial resolution for our analysis. We preferred ERA5 because it made the analysis quicker and easier as CARRA is a relatively heavy data set on a less convenient grid.

Line 120-121: 'the multitaper method from Percival and Walden (1993) with Slepian data tapers (e.g., Slepian, 1978; Thomson, 1982). To analyse the time-variability of the spectral components, wavelet transforms following Lilly and Olhede (2009) with generalised Morse wavelets (Olhede and Walden, 2002) were used' – adding a brief explanation about the general purpose of using the methods mentioned (rationale behind and how they contribute to the study's objectives) in the manuscript could enhance the reader's understanding and contribute to smoother reading.

We agree. Our general purpose in using these methods was to identify different time scales associated with different processes. To clarify this, we changed the first sentence to "We identify the time scales of variability using spectral analysis" (Line 120), and reformulated the last sentence to "The values for the parameters that define the Morse wavelets were chosen to be gamma = 3 for symmetric wavelets and beta = 5 to resolve the variability on time scales between semidiurnal and a few weeks in both frequency and time." (Lines 123–125)

Line 134: 'bottom half of the water column' – why? Do you aim to separate the surface layer circulation or rather ADCP data quality away from transducer was a limiting factor here?

The ADCP data quality was good through most of the water column throughout the time series (the range shown in Figure 6d is the range with quality-controlled data). We chose to use the layer-average over the bottom half of the water column because temperature measurements were only available near the bottom. Available hydrographic profiles indicate that although the water column is at times stratified, the lower half of the water column is mixed enough for the near-bottom temperature to be representative.

We clarified this by adding "The velocity data are layer-averaged over the bottom half of the water column. While the water column is at times stratified in temperature, available CTD profiles suggest that the lower half of the water column is typically well mixed thus temperature measured close to the bottom may be regarded to be representative for the lower half of the water column." (Lines 144–147)

*Results:*

I advise the authors to consider prioritising the most significant results and simplifying the presentation of detailed analyses, particularly in Section 3.4. The abundance of details makes it challenging for readers to follow the discussion effectively. Instead of presenting all available analyses, the authors might consider using a simple table with numerical data to present these results. Additionally, referring large part of the results solely to Figure 8 can be difficult for readers.

Thank you for the constructive feedback. We agree that there are details in the text and figures (especially in Section 3.4 and Figure 8), which are not critical for our results and conclusions. We do agree that especially Section 3.4 and Figure 8 can be simplified. In the revised version we removed Figure 8, as the important parts are visible in other figures (Figures 7, 9, and 10), and shortened the text in Section 3.4 by approximately three paragraphs.

Line 152: 'salinity has increased by 0.05 g/kg to 0.35g/kg'- it's interesting how 34.4 isohaline deepened, and that negative T anomaly is noticeable in this location as well (Fig 4b-c) in the last decades in the Olga Basin. Is it an effect of more intense mixing? Or perhaps this is a signal of the Polar Front sharpening and concurrent steepening of isohalines/isopycnals? As we know from 'Introduction' section (Lines 38-42) the amount of inflowing freshwater has decreased in the northern Barents Sea…

Thank you for pointing this out. We now mention the deepening of the 34.4 g/kg isohaline toward the Olga Basin and the colocated negative T anomaly (lines 168–170). In order to answer your question, a targeted analysis is needed. Having sparse data in the Olga Basin, we are hesitant to interpret "details" or areas with especially sparse coverage and low standard deviation (due to coverage). In Section 4.4, we have a short discussion on the role of changing stratification in the Olga Basin and included the sentence "We noted the deepening of the 34.4 g kg−1 isobath and a small decrease in temperature in parts of the Olga Basin (Fig. 4b–c) in the recent two decades. However, due to the particularly sparse data coverage in the Olga Basin, we are cautious about interpreting these changes." in lines 442–444 in this section.

Line 161: 'At this time, AW was present in Storfjordrenna and close to the mooring' – again, I miss the water mass classification in this manuscript. As the AW definition was not mentioned in the Introduction and Data & methods part, it's necessary to introduce it here before you start to describe the results. As you use CT and SA, it will be good to define AW in these scales.

We agree. AW and Atlantic-origin water are defined in the Data and Methods section (see above).

We also specified the AW properties observed in Storfjordrenna on this transect in the sentence "At this time, AW was present in Storfjordrenna and close to the mooring (not shown): 70 km southwest of the mooring, water at 2.9 ◦C to 3.7 ◦C and 34.84 g/kg to 35.02 g/kg was found at depth, which is within the temperature range for pure AW and slightly lower than the salinity threshold of 35.06 g/kg (Sundfjord et al., 2020). Also relatively warm water of around 0.5 ◦C was situated at depth 10 km to the west of the mooring position."

Line 198: 'to 34 cm s−1 near the surface' - when exactly? It will be good to find it in the plot - one cannot see the local maxima near the surface.

Thank you for pointing this out. It was in the beginning of March, and "near the surface" was misleading. We have reformulated this sentence to read "reaching 34 cm s−1 at the beginning of March, the highest value measured during the whole sampling period (Fig. 6c–d)" (lines 211–212).

*Discussion:*

Winds, ice, tides, and AW inflows as well as upstream conditions definitely work together to shape the seasonal and interannual variability. However, attempting to address all the complex factors in one comprehensive article is probably ambitious task. I found the discussion too broad, it's understandable given the extensive range of results presented in previous sections. However, I miss a clear statement regarding which mechanisms are deemed most responsible for the observed high interannual variability.

We agree that there are many complex factors shaping the observed seasonal and interannual variability. It is clear that we cannot explain everything with the data at hand, but we argue that our analyses in conjunction with information from the literature can provide important information about the likely drivers of variability. In order to address the reviewers valid concerns, we aimed to streamline the discussion by putting it more directly in the context of recent literature, as well as improving the clarity of our concluding statements:

Early in Section 4.1, we added a reference to Lundesgaard et al. (2022) who observed a similar year-to-year difference in the seasonal cycle in the Northern Barents Sea Opening and a reference to Lind et al. (2018) about the preconditioning of imported sea ice melt on the local sea ice growth the following winter:

"During the same autumns, observations in the Northern Barents Sea Opening showed a large difference in the upper ocean salinity accompanied by a similar difference in sea ice (Lundesgaard et al., 2022). Higher upper ocean freshening due to more sea ice melt can precondition the area for sea ice growth the following winter (Lind et al., 2018)." (Lines 298–300)

We have edited the last paragraph of Section 4.1 and included a couple of references about interannual variability in the Barents Sea and its drivers:

"To summarise, strong interannual variability substantially impacts the seasonal cycle at the study site. Sea ice in the Barents Sea has a large interannual variability (e.g., Shi et al., 2024; Onarheim et al., 2024) driven by atmospheric temperature, sea surface temperature, and oceanic heat transport (Dörr et al., 2024). Locally, intrusions of warm water during autumn can delay the onset of winter conditions and inhibit local sea ice growth or melt imported sea ice, as observed in the autumns of 2018 and 2020 (Fig. 6b–c and 5b). These intrusions, whether driven by local winds or by upstream conditions, also frequently reduce the existing partial ice cover above the saddle during winter and spring (Fig. 11, earlier years not shown). This explains the differences in hydrographic conditions observed in November 2019 vs 2021

(Fig. 5a and 5c). The more persistent warm water overflow in 2020 likely delayed the seasonal watermass transformation and contributed to the October 2020 transect being substantially warmer than the November transects." (Lines 341–349)

In Conclusions, we added that weather-band processes lead about twice as much heat across the saddle as the semidiurnal tide (Lines 459–461) and added the sentence "The complex interplay of local and non-local processes governing the large variability of AW inflow on seasonal to interannual timescales warrants more comprehensive data collection and analysis in this area." (Lines 467–469)

Line 323: 'water column has not yet cooled down at that time' – indeed, the observed differences between mid-October and November suggest a potential regime shift from Atlantic Water (AW) dominance to Arctic Water (AW) dominance during that time of the year. But of course, it could also be an exceptional year with more/warmer inflowing AW as you wrote.

Fig. 5 indeed indicates a shift from Atlantic to Arctic Water dominance during that time of year due to atmospheric cooling and influence from the colder, fresher environment in the northeast. However, our Fig. 12 suggests that for autumn 2020, there was a stronger inflow of Atlantic Water lasting into winter, unlike other years where November was either Arctic dominated or cooled down. Autumn 2018 stood out as warmer and saltier through October and November, with sea ice starting later (Figure 6). Lundesgaard et al. (2022) also observed this south of Kvitøya. We interpret this as, in addition to the normal seasonal cycle, interannual variability in Atlantic Water inflow to the northern Barents Sea substantially influences the seasonal transformation in the study area. We modified the discussion in the manuscript to better reflect this point and included some references:

"The November transects (Fig. 5a,c) were both cooler and less saline than the October transect (Fig. 5b). This may be due to a regular seasonal shift from AW influence to Arctic Water influence at the site and/or interannual variability in this region." (Lines 326–327)

"Atlantic inflow from the north of Svalbard has also been observed to affect the seasonal variability in local hydrography in the same way (Lundesgaard et al., 2022). (Lines 330–332)

"To summarise, strong interannual variability substantially impacts the seasonal cycle at the study site. Sea ice in the Barents Sea has a large interannual variability (e.g., Shi et al., 2024; Onarheim et al., 2024) driven by atmospheric temperature, sea surface temperature, and oceanic heat transport (Dörr et al., 2024)." (Lines 341–343)

Figures:

In general, I think a few plots and subplots would be better with some additional information that would make them more self-explanatory. Increasing the thickness of lines, enlarging markers, and reducing clutter can improve the visibility of key data points. It feels like this manuscript has many more figures than only 12 (plus these in appendices).

Figure 3: isotherm 0 deg C (light grey) - too like 200 m isobath - can it be changed?

We agree and changed the colour of the isotherm in Figure 3 as well as the isohalines in Figures 3, 4, and 5 to ensure that they are all visible and distinguishable from each other. Isohalines are pale blue with darker edges to stand out from all colours in the temperature colourmap, and the 0 degree C isotherm is bright blue. We also made the 200 m isobath brighter.

Figure 7f-g: Perhaps some simple legend naming the lines can be added to read these plots without checking back and forward with the caption?

We have added a legend and also given each curve a distinct colour. We now show both rotary components for both seasons in a shared panel to reduce the number of curves.

Figure 8: This plot seems a bit too complicated to me. Also, it would profit from adding some legends.

We have removed Figure 8 since it contained information that was not central to the manuscript. We agree with your suggestion above to prioritise the most significant results. The main results which we wanted to communicate through Figure 8 are also visible in Figures 7, 9, and 10. Furthermore, removing Figure 8 allows removing some text that describes this figure, and further simplifies the content for the reader.

Figure 9: Why is there no current anomalies arrow in the subplot c)? The reference unit vectors for wind and currents are missing, as well.

Thank you for noticing. We have added reference vectors for wind stress (0.2 W/m^2) in panel b and for current (5 cm/a) in panel d.

In Fig. 9c, the composite average of the current anomaly is minuscule (about a hundredth of the current anomalies in panels b and d) because we have averaged over the non-extreme wind stress conditions. Panel c shows the spatial picture of the centre of the cross (standard deviation cross) for the pale blue dots in panel a, which represents weak wind stress from north/northeast (small negative Ekman transport) and a (very close to) zero current anomaly.

Figure 12: 'Sea ice concentration (black, left vertical axis) and sea surface temperature (red, right vertical axis)' - from which product - write it here, also - add years on the subplots - it would be easier to look at.

Thank you for this suggestion. We have added the years on the subplots and added that the data are from OSTIA.

**Response to Reviewer 2's comments**

In the presented manuscript the Authors propose a future pathway of Atlantic Water towards the Barents Sea, taking place across the channel between Edgeøya and Hopen islands. As it appears to be smaller than other existing AW pathways, the Authors emphasize the role of this inflow on the Olga Basin, reservoir of the coldest and thickest Arctic Water layer, and impact on East Spitsbergen Current important for the shelf and fjords west of Svalbard. Under this assumption, the Authors provide a comprehensive analysis of interannual and seasonal variability of hydrographic conditions in the region based on historical and new data during late autumn as well as year-long data from mooring.

The paper addresses important aspects of the Barents Sea region, being a hot-spot in Arctic picture and falls very well in the scope of OS. The data are novel and provide crucial information on largely unexplored region. The study site description is comprehensive, data and methods are carefully selected to extract as much information as possible to provide valuable results and discussion. However, sometimes it appears to be overloaded and difficult to receive, and some things need to be rephrased/restructured to reach a broader audience. Nevertheless, I believe that this paper is valuable for scientific community and can serve as an important input to understanding the Atlantification of the Barents Sea. Below, I suggest some minor corrections before publication

Thank you for your review. Your constructive feedback helped improving our manuscript substantially. In the revised version, we have improved clarity and readability. Please see below where we address each comment.

**Introduction**

I find the Introduction very accessible, readable and comprehensive. The only suggestion is to rephrase/create the hypothesis or state research questions to make the goal of the paper more emphasized. From Introduction it appears that this small area (the channel between Edgeøya and Hopen islands) is largely unexplored, which makes your data novel and of great importance. We also know, that AW in Storfjordrenna is becoming warmer and is observed at shallower depths, so first you can hypothesise that due to the above observations the expansion of AW through the channel may increase leading to the formation of a future pathway of AW (persistent?) into the Arctic Ocean. Then, you can write about the role of this inflow, e.g. the sentence in line 72 and I think it is also worth mentioning, that increased AW inflow here can be of great importance for ESC (you mentioned it in Discussion: line 462), which on the other hand, is crucial for the marine and coastal environment of west Spitsbergen.

We are pleased to hear that the Introduction was positively received, and appreciate your suggestions to emphasise the hypothesis and research goal. In response, we have incorporated comments in the Introduction on the effects of AW inflow on sea ice and stratification in the Olga Basin, the ESC, and the environment in Storfjorden and along the west of Spitsbergen. The last paragraph now reads:

"The channel between Edgeøya and Hopen islands (Fig. 2) separates the increasingly warm and shoaling AW in Storfjordrenna from the Olga Basin, the Arctic domain in the northwestern Barents Sea. Here, we investigate the physical processes that drive and mediate the inflow of Atlantic-origin water masses into the Olga Basin. Such warm water inflow can potentially reduce growth, accelerate sea ice melt and change the deep basin stratification in the Olga Basin. The (increasing) presence of warm Atlantic-origin waters in Storfjordrenna may also have implications for the properties of the East Spitsbergen Current crossing the channel westward, Storfjorden, and the coastal environment west of Spitsbergen. We propose that the study site may be an increasingly relevant pathway for warm waters into the Arctic domain of the Barents Sea. Using novel data from a less-explored region, we identify and discuss the variability of the currents and heat exchange on time scales from hours to months." (Lines 68–76)

Figure 2: Red dots across the saddle are not clearly visible, which makes wrong first impression, that the transect in 2019 is only made along the channel. Please fix this. Are the dots representing the transect from Storfjordrenna to Olga Basin the same as the transect shown in Figure 3?

Thank you for pointing this out – we agree that this should be clearer. We have increased the size of the dots along and across the saddle and adjusted the colours to make them more distinguishable from each other. The transect indicated by the red dots in Figure 2 is not exactly the same as the transect marked in Figure 3. To facilitate easier comparison, we have added dots to indicate the climatological section in Figure 2 as well.

**Data and methods**

This section includes a lot of details, which is good and proves that the Authors carefully and thoughtfully selected data and analysis methods to defined scientific problems. However, the description, despite specific information, makes this section available for rather narrow audience. This is especially true for mooring data. To clarify, I suggest stating clearly at the beginning of the paragraph in line 114, what the mooring data are used for, that those data are utilized to investigate variability in ocean currents and temperature associated with a) let's say direction of the flow, b) selected time scales, including semidiurnal tides, weather-band processes (explanation/example?) and low-frequency activity (explanation/example?). Therefore, for a) the coordinate system was rotated with -42°… and for b) -28°… . This small rephrasing will help more clearly see and remember how the data will be analyzed. Then you can write about cutoff frequencies for different time scales.

Thank you for this suggestion. We have revised this part along the lines you suggest. We now introduce this paragraph with: "We analyse the current velocity data from the mooring to study the mean flow properties and variability of the flow in time scales associated with tidal variability, weather-band processes, and lower-frequency events. For this we used two different coordinate system rotations." (Lines 133–135)

Line 120-121: I think this needs a little explanation of the method and the purpose/justification for its application.

We have included a sentence in the beginning of this paragraph to clarify the purpose: "We identify the time scales of variability using spectral analysis" (Line 120), and reformulated the last sentence to "The values for the parameters that define the Morse wavelets were chosen to be gamma = 3 for symmetric wavelets and beta = 5 to resolve the variability on time scales between semidiurnal and a few weeks in both frequency and time." (Lines 123–125)

**Results**

Lines 148-149: What is the reason for choosing this particular isoline (34.7 g kg-1)? Can it be representative for "pure" or Atlantic origin water? As you mentioned in the Introduction, that you investigate the Atlantic origin inflow in this area, it would be useful to define this somehow and follow the idea.

The salinity of this isohaline is slightly too low to represent pure AW. If the study site had been more dominated by pure AW, using the salinity threshold for AW would have been more informative. Instead, we chose a less saline isohaline representing a mixture of AW with fresher ambient waters, illustrating how this Atlantic-origin water has expanded. For further details about the water mass definitions, please refer to the comment below.

Line 151: Could you more specify Atlantic origin water? I understand that this is rather challenging, especially when you describe seasonal evolution, where water undergoes significant modification and AW just provides heat and salt to the region. However,  maybe it would be appropriate to define it/add some description (one sentence) in the Introduction? Or when you say about, for example, "pure" AW, then in brackets you can give some values stating this.

We agree. In the Data and Methods section, we have added a definition of Atlantic Water (Conservative Temperature exceeding 2 degrees and Absolute Salinity exceeding 35.06 g/kg following Sundfjord et al., 2020) and also clarified what we mean by "Atlantic-origin water" by adding the sentence "We refer to the Atlantic-origin water as the AW that is modified or transformed en route from the WSC through Storfjordrenna into relatively warm and saline water compared to the surrounding water masses, but colder and less saline than pure AW." (Lines 117–119)

Line 155-157: Figure 5 says that all measurements (Nov 2019, Oct 2020, Nov 2021) were performed across the saddle. However, from Figure 2 I kept in mind (I was pretty sure), that Nov 2019 was preformed from Storfjorden to Olga Basin (red dots). As I mentioned red dots across the saddle in Figure 2 are not clearly visible, which makes a wrong first impression. Please fix this.

We agree that this was not clear enough. We have adjusted the size and colours of the transect markers in Figure 2. It is now visible that the transect from Edgeøya to Hopen was done in all three years.

Line 176: "close to the pure AW properties" means what exactly? This water hasn't been defined in the manuscript before.

We have added the water mass definition for pure AW in the Data and Methods section and further clarified the temperature and salinity observed here. Lines 175–178 now read:

"70 km southwest of the mooring, water at 2.9 °C to 3.7 °C and 34.84 g/kg to 35.02 g/kg was found at depth, which is within the temperature range for pure AW and slightly lower than the salinity threshold of 35.06 g/kg (Sundfjord et al., 2020). Also relatively warm water of around 0.5 °C was situated at depth 10 km to the west of the mooring position."

Line 258: Here you give some examples of analyzed time scale, which should be clearly stated in section Data and methods.

We have clarified in the Data and Methods section how we identified these time scales by modifying the paragraph about band-pass filtering:

"The spectral analysis revealed three frequency bands with dominant variability, which we associate with semidiurnal tides (periods of ten hours to 14 hours), "weather-band" processes (28 hours to 10 days), and lower-frequency activity (seven days to six weeks). To study these fluctuations, we filtered the time series with a Butterworth band-pass filter with an order as high as possible while ensuring stability for each frequency band, order 5 for semidiurnal tides and weather-band, and order 3 for lower-frequency activity."

Figure 7: Dotted lines representing cutoff frequencies for the semidiurnal tidal band are hardly seen. Is it possible to make them more visible?

We increased the size of all dotted lines for the semidiurnal tidal band.

**Discussion**

This section is also overloaded with information and sometimes it is hard to follow. I propose to make some corrections for better reading of the text.

In section 4.1 you want to discuss interannual and seasonal variability in hydrography of the region with asking a question if 2018 – 2019 was a typical year. I suggest to start from discussing only a mooring year, comparing autumns 2018 and 2019, explaining AW inflows. Then you can make a comparison with other years (transect across the saddle and time series from Figure 12). At the end you can address the question asked.

Thank you for the suggestion. We agree that it improves the flow of the section. In the revised manuscript, we have reorganised Section 4.1 to begin with comparing the autumns of 2018 and 2019. We then describe and discuss the warm water intrusions in the mooring year, followed by a discussion on the transects on the saddle together with the time series in Figure 12, before finally addressing the main question. In addition, some paragraphs were condensed for brevity.

Line 333: "However, the mean SST in 2018–2019 was higher than all years between 1981 and 2006" - Does it add anything significant to the discussion?

Thank you for pointing this out – it is indeed not significant to this section of the discussion. The intent was to show that 2018–2019 was a "normal" or relatively average year in the context of the new warmer era. We moved this sentence and much of the rest of the paragraph to Section 4.4, where it fits better.

Figure 12: Please add to the figure caption, that this is from the mooring location.

Thank you – we have adjusted the caption.

Figure A1 and A2: please add to the captions what the magenta diamond indicates.

We added "The mooring location is indicated with a pink diamond."